# Enhancing Domain Adaptation through Prompt Gradient Alignment

**Hoang Phan**[*]
New York University
hvp2011@nyu.edu

**Lam Tran**[*]
VinAI Research
lamtt12@vinai.io

**Quyen Tran**[*]
VinAI Research
quyentt15@vinai.io

**Trung Le**
Monash University
trunglm@monash.edu

## Abstract

Prior Unsupervised Domain Adaptation (UDA) methods often aim to train a domain-invariant feature extractor, which may hinder the model from learning sufficiently discriminative features. To tackle this, a line of works based on prompt learning leverages the power of large-scale pre-trained vision-language models to learn both domain-invariant and specific features through a set of domain-agnostic and domain-specific learnable prompts. Those studies typically enforce invariant constraints on representation, output, or prompt space to learn such prompts. Differently, we cast UDA as a multiple-objective optimization problem in which each objective is represented by a domain loss. Under this new framework, we propose aligning per-objective gradients to foster consensus between them. Additionally, to prevent potential overfitting when fine-tuning this deep learning architecture, we penalize the norm of these gradients. To achieve these goals, we devise a practical gradient update procedure that can work under both single-source and multi-source UDA. Empirically, our method consistently surpasses other vision language model adaptation methods by a large margin on a wide range of benchmarks. The implementation is available at `https://github.com/VietHoang1512/PGA`.

## 1 Introduction

Deep learning has significantly advanced the field of computer vision, achieving remarkable performance in tasks such as image classification [1–5], object detection [6–9], and semantic segmentation [10–13]. However, the effectiveness of these deep learning models heavily relies on large amounts of labeled training data, which is often labor-intensive and expensive to collect. Moreover, the discrepancy between training data and real-world testing data can lead to substantial performance drops when models are deployed in practical settings [14–16].

To address these challenges, Unsupervised Domain Adaptation (UDA) has emerged as a pivotal solution. UDA aims to transfer knowledge from a labeled source domain to an unlabeled target domain in the presence of a domain shift, thereby enabling models to generalize well across different domains without requiring extensive labeled data for the target domain. This is often achieved by optimizing objective function on source domains and other auxiliary terms that encourage learning domain-invariant feature representations [17–20] or enhance model robustness [21–24], which mitigates the domain shift and improve the performance on unseen data. Nevertheless, aligning representations could potentially hurt the model performance due to the loss of discriminative features [25, 26].

---

[*]Equal contributions.

38th Conference on Neural Information Processing Systems (NeurIPS 2024).

Conceptually, our proposed method is orthogonal to these invariant feature learning methods, and they could complement each other.

Recent works leveraging pre-trained models like CLIP [27] for UDA can significantly bridge domain gaps and improve generalization by utilizing rich semantic information and robust visual representations through extensive pre-training on diverse image-text datasets. Following this vein, DAPL [25] first introduces domain-specific and domain-agnostic prompts to efficiently adapt pre-trained vision-language models without fine-tuning the entire model. Furthermore, MPA [28] aligns multiple prompts from different sources using an auto-encoder. While these methods could obtain superior performance on different benchmarks, we find that the main improvement comes from the strong zero-shot performance and self-training mechanism. In particular, prior works often generate pseudo-label for unlabeled images and then train the model on those samples. Consequently, finetuning a pretrained CLIP model on this dataset alone without leveraging source datasets can help refine model prediction significantly, boosting the performance from $88.1\%$ to $90.1\%$, yielding a competitive result compared against MPA, as presented in Table 1.

| Dataset | $\rightarrow$ C | $\rightarrow$ I | $\rightarrow$ P | Avg |
|---|---|---|---|---|
| Zero-shot | 87.9 | 88.2 | 78.7 | 88.1 |
| Simple Prompt | 93.6 | 90.6 | 80.9 | 88.4 |
| Self-training | 92.9 | 94.3 | 83.2 | 90.1 |
| MPA | 97.2 | 96.2 | 80.4 | 91.3 |

Table 1: Self-training on pseudo-labeled target data is already a strong baseline.

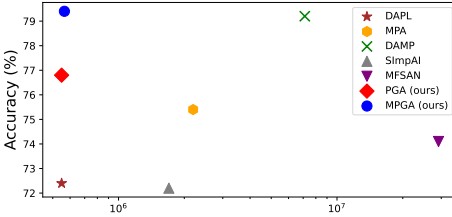

Figure 1: Baselines performance on Office-Home

Motivated by this observation, we directly optimize the main objective function not only on source domains but also on the target data, instead of only using them for auxiliary objectives as in previous work [29, 30]. We thus cast the original UDA problem as a multi-objective optimization (MOO) problem. Specifically, we minimize a vector-valued loss function, which includes the objectives of multiple source domains and the target domain. This formulation allows us to apply existing results from MOO literature for finding Pareto solutions, from which we can not optimize an objective without hurting another [31–33] or encourage positive inter-task transfer between objectives [34–37]. Note that in the context of UDA, we focus more on learning the target task, thus motivating us to apply prioritized MOO algorithms [38–40] or to incorporate predefined preferences [32, 41–43]. While those methods allow practitioners to focus more or less on the objectives at hand, they come with the cost of extensive hyperparameter tuning. Besides, recent works [44–46] argue that simple loss functions reweighting can match the performance of gradient-based MOO methods [35, 47]. Those findings suggest we focus more on the inherent conflict nature of per-objective gradients instead of attempting to remove the conflict between them [35, 34].

In this paper, we propose casting the problem of UDA as a multi-objective optimization by leveraging powerful pre-trained models. However, while obtaining impressive results on various downstream tasks, over-parameterization is still a crucial problem for transformer-based models [48, 49], which potentially causes overfitting [50–54] more severely than small-parameter architectures [55], especially in the multi-task learning context. For that reason, we propose to (i) fine-tune pre-trained model via prompt learning, which is known for being more robust [56–58] and especially more light-weight than full fine-tuning, (ii) and to leverage the gradient norm penalty to encourage model generalization [59, 24, 60, 61]. Furthermore, we introduce a gradient alignment algorithm to foster inherent consensus between per-objective gradients without modifying the gradient itself. Our proposed method, termed Prompt Gradient Alignment (PGA), and its variant for multi-source UDA, Multi-Prompt Gradient Alignment (MPGA), achieve state-of-the-art performance on different UDA benchmarks. As shown in Figure 1, PGA and MPGA outperform traditional UDA methods like MFSAN [62] and recent prompt-based UDA methods such as MPA [28] and DAMP [63] while requiring fewer trainable parameters. We also provide a generalization bound for UDA and show how theoretical insights motivate the design of our proposed method.

## 2 Related work

**Unsupervised Domain Adaptation.** A dominant approach to solving the UDA problem is to reduce the distribution shift between source and target domains. Following the foundational theory outlined in [64], one group of methods seeks to minimize the H-divergence between the marginal distributions of these domains [65–67]. Alternatively, other methods aim to align the moments of these distributions, as suggested in [68–70]. Additionally, adversarial learning has been employed to learn domain-invariant features. For instance, methods such as those in [19, 71] use a domain discriminator to differentiate between source and target samples, training a feature extractor to deceive this discriminator. However, as [25] highlights, these methods often struggle with a trade-off between domain alignment and classification performance, particularly in multi-source scenarios where only a single model is used.

**Prompt learning-based domain adaptation** is a novel approach introduced in [25], leverages the generalization capabilities of CLIP to learn both domain-agnostic and domain-specific prompts. This method effectively addresses the trade-off between domain alignment and classification performance by employing a contrastive loss. This loss aligns the representation of an image with the prompt corresponding to its ground truth class and domain, thereby encouraging the learning of domain-invariant features. Building on this foundation, MPA [28] advances the concept of multi-source UDA. It adapts the prompt learning strategy to each source-target domain pair. The prompts are aligned through a denoising auto-encoder using Euclidean distance. However, prompting is known as a brittle process where a small shift to the prompt can cause large variations in the model predictions [72–74]. Therefore, in this work, we propose to intervene in the training on the gradient space as it offers a more interpretable and controllable effect during training. Furthermore, PGA is trained in an end-to-end fashion, avoiding the sequential training for each source-target pair as in MPA.

**Gradient-based multi-task learning.** Due to the multi-objective nature of the multi-source domain adaptation problem, one can leverage recent methods from the multi-task learning literature [34, 35, 75] to derive an optimization procedure that benefits the learning across domains or put more weight on some specific domains via incorporating preference [32, 41, 42]. While those techniques are readily applicable in our context, we directly re-weight per-task gradients, similar to scalarization, instead of adopting multi-task learning methods for simplicity. Furthermore, our work is orthogonal to those gradient-based multi-task learning methods where we encourage the consensus among objects instead of directly manipulating their gradients to remove inherent conflicts among them.

**Gradient matching** is commonly used in continual learning [76–78] to measure conflict and transferability between tasks. A positive dot product between two tasks' gradients implies updating the models with one task can benefit the other. This principle is also applied in domain generalization [79, 80] to focus on invariant features. However, our approach aligns in the space of prompt gradient, a significantly smaller parameter set than the full model gradients used in previous works. Besides, to avoid the computation of costly second-order derivatives, [79] linearly approximate the inner product between gradients, which underperforms on datasets with a larger number of domains due to cumulative approximation error. Meanwhile, our method does not face this problem since we implicitly compute this term without using any approximation. More works sharing the same intuition of gradient alignment are provided in Appendix D.

## 3 Background

### 3.1 Unsupervised Domain Adaptation

Given a set of $N \geq 1$ source domains $\{D_{S,i}\}_{i=1}^{N}$ each of which is a collection of data-label pairs of domain $i$, i.e. $D_{S,i} = \{x_j, y_j\}_{j=1}^{N_{S,i}}$, and one unlabelled target domain $D_T = \{x_j\}_{j=1}^{N_T}$, where $N_{S,i}$ and $N_T$ are respectively the number of data points in source domain $i$ and target domain $T$, the goal is to learn a model that can perform well on the unlabelled target domain. In this paper, we focus on classification problems and denote $K$ as the number of categories.

### 3.2 Prompt Learning on CLIP-based models

CLIP [27] is a vision-language model that consists of an image encoder $f_i$ and a text encoder $f_t$, which is trained to align the visual representation $f_i(x)$ of an image $x$ with the textual representation

$f_t(y)$ of the corresponding label. The textual representation is derived from a manually crafted prompt $\boldsymbol{p}_k$ in the form "A photo of a [CLASS]$_k$", where [CLASS]$_k$ is the class $k$'s name. With great generalization capability, pre-trained CLIP models are often used for a variety of downstream tasks through prompt learning.

For zero-shot inference, $K$ class names are forwarded through the text encoder, and the one with the highest representation similarity with the image is the predicted class:

$$y_{\text{pred}} = \operatorname{argmax}_k P(y = k | \boldsymbol{x}), \text{ where } P(y = k | \boldsymbol{x}) = \frac{\exp(\langle f_i(\boldsymbol{x}), f_t(\boldsymbol{p}_k) \rangle / \gamma)}{\sum_{k'=1}^{K} \exp(\langle f_i(\boldsymbol{x}), f_t(\boldsymbol{p}_{k'}) \rangle / \gamma)}, \quad (1)$$

and $\langle ., . \rangle$ measures the cosine similarity and $\gamma$ is the temperature.

For fine-tuning, a set of learnable class-shared prompts are added to the class token to form $\boldsymbol{P}_k = [\boldsymbol{v}_1 | \boldsymbol{v}_2 | \cdots | \boldsymbol{v}_M][\text{CLASS}]_k$, where $\boldsymbol{v}_i$ is a vector with the same size as the word embedding, and $M$ is the number of added prompts. These prompts are learnt by maximizing log-likelihood on downstream data, i.e. $\max \sum_i \log P(y = y_i | \boldsymbol{x}_i, \boldsymbol{P})$. Note that in this predictive probability, we abuse symbol $\boldsymbol{P}$ to refer to the learnable tokens $\boldsymbol{v}_i$, and when we drop the symbol as in 1, we refer to the zero-shot prediction using CLIP. As a result, additional information about the downstream task can be encoded in the prompts, and this design will enable knowledge transfer from pre-trained datasets.

# 4 Proposed method

In this section, we describe our proposed prompt gradient alignment method. Motivated by the lightweight and effective nature of prompt learning in adapting pre-trained knowledge to downstream tasks, we cast UDA as a multi-objective optimization (MOO) problem, from which we propose aligning gradients of different objectives and minimizing their norms simultaneously. Additionally, we derive a UDA generalization bound to justify the intuition of our method. The full details of our proposed method in the generalized case where we have more than one source domain are provided in Appendix B.

## 4.1 Prompt design

A common assumption in domain adaptation literature is that each domain can be represented by domain-specific features and those that are shared with others. To reflect this, we employ two sets of prompts for each domain: domain-agnostic prompt (or shared prompt, interchangeably) $\boldsymbol{P}_{sh}$, and domain-specific prompts $\boldsymbol{P}_{S,i}$ and $\boldsymbol{P}_T$. Here, $\boldsymbol{P}_{S,i}$ refers to prompt used for source domain $i$, and $\boldsymbol{P}_T$ is that for target one. In particular, following DAPL, we use $K \times M_1$ tokens to construct $\boldsymbol{P}_{sh} = [\boldsymbol{P}_{sh}^k]_{k=1}^K$, where $\boldsymbol{P}_{sh}^k = [\boldsymbol{v}_1^k | \boldsymbol{v}_2^k | \cdots | \boldsymbol{v}_{M_1}^k]$ is class-specific shared tokens. For source- and target-specific prompts, we use $M_2$ tokens: $\boldsymbol{P}_{S,i} = [\boldsymbol{u}_1^{S,i} | \boldsymbol{u}_2^{S,i} | \cdots | \boldsymbol{u}_{M_2}^{S,i}]$, $\boldsymbol{P}_T = [\boldsymbol{u}_1^T | \boldsymbol{u}_2^T | \cdots | \boldsymbol{u}_{M_2}^T]$. And denote $\boldsymbol{P} = [\boldsymbol{P}_{sh}, \{\boldsymbol{P}_{S,i}\}_{i=1}^N, \boldsymbol{P}_T]$ as the whole prompts used in our method. Based on this, we use a prompt of the form $[\boldsymbol{P}_{sh}^k][\boldsymbol{P}_{S,i}][\text{CLASS}]_k$ to compute the predictive distribution of a source $i$ sample belonging to class $k$, and similarly $[\boldsymbol{P}_{sh}^k][\boldsymbol{P}_T][\text{CLASS}]_k$ for a target sample.

## 4.2 Empirical risk minimization: a simple baseline

As we introduced, to learn those prompts, we consider the cross-entropy losses applied to source data and target data with pseudo labels as a set of objectives to optimize simultaneously:

$$\mathcal{L}_{total}(\boldsymbol{P}) := \left[ [\mathcal{L}_{S,i}(\boldsymbol{P})]_{i=1}^N, \mathcal{L}_T(\boldsymbol{P}) \right] = \left[ [\mathcal{L}_{S,i}(\boldsymbol{P}_{sh}, \boldsymbol{P}_{S,i})]_{i=1}^N, \mathcal{L}_T(\boldsymbol{P}_{sh}, \boldsymbol{P}_T) \right],$$

$$\mathcal{L}_{S,i}(\boldsymbol{P}_{sh}, \boldsymbol{P}_{S,i}) = \text{CE}(\boldsymbol{P}_{sh}, \boldsymbol{P}_{S,i}; \boldsymbol{X}_{S,i}, \boldsymbol{Y}_{S,i}) = -\frac{1}{N_{S,i}} \sum_{j=1}^{N_{S,i}} \log P(y = y_j | \boldsymbol{x}_j, \boldsymbol{P}_{sh}, \boldsymbol{P}_{S,i}), \quad (2)$$

$$\mathcal{L}_T(\boldsymbol{P}_{sh}, \boldsymbol{P}_T) = \text{CE}_\tau(\boldsymbol{P}_{sh}, \boldsymbol{P}_T; \boldsymbol{X}_T, \boldsymbol{Y}_T)$$

$$= -\frac{1}{N_T} \sum_{j=1}^{N_T} \mathbb{I}(P(y = \hat{y}_j | \boldsymbol{x}_j) \geq \tau) \log P(y = \hat{y}_j | \boldsymbol{x}_j, \boldsymbol{P}_{sh}, \boldsymbol{P}_T), \quad (3)$$

$$\hat{y}_j = \arg\max_k P(y = k | \boldsymbol{x}_j). \quad (4)$$

In summary, the total loss consists of $N + 1$ objectives. The target objective is applied to target samples whose zero-shot predictions made by CLIP are larger than a threshold $\tau$.

Given these objectives, source- and target-specific prompts can be updated by minimizing source and target losses, respectively. Regarding domain-agnostic prompt, one can put a weighting term on the signal from source losses to compute the gradient. Formally, for $\forall i = 1 \rightarrow N$, we have:

$$\boldsymbol{g}_{sh,i}, \boldsymbol{g}_{S,i} = \nabla_{\boldsymbol{P}}\mathcal{L}_{S,i}(\boldsymbol{P}_{sh}, \boldsymbol{P}_{S,i}), \quad \boldsymbol{g}_{sh,T}, \boldsymbol{g}_T = \nabla_{\boldsymbol{P}}\mathcal{L}_T(\boldsymbol{P}_{sh}, \boldsymbol{P}_T),$$
$$\boldsymbol{P}_{S,i} = \boldsymbol{P}_{S,i} - \eta\boldsymbol{g}_{S,i}, \qquad\qquad \boldsymbol{P}_T = \boldsymbol{P}_T - \eta\boldsymbol{g}_T, \tag{5}$$
$$\boldsymbol{P}_{sh} = \boldsymbol{P}_{sh} - \eta(\boldsymbol{g}_{sh,T} + \lambda\sum_i \boldsymbol{g}_{sh,i}), \tag{6}$$

where $\eta$ is the learning rate, and $\lambda$ is the weighting term to control how much emphasis we want to put on the target domain. Note that we treat gradient signals from source domains equally as we assume no prior preference knowledge about them. Nevertheless, one can measure the domain similarity between each source and target domain to devise a better way to reweight source domains' objectives. However, as will be shown in the experiments, taking the average is simple yet yields superior results, hence we will leave this for future work.

## 4.3 Prompt gradient alignment for UDA

For simplicity, we first consider the single-source UDA setting and will present the extension to the multi-source one later in Appendix B. One problem with the method above is we ignored the potential inherent gradient conflict between objectives when updating the shared prompt. To mitigate this, one can follow gradient-based methods, such as [35, 47] to manipulate the gradients so that conflict is reduced. However, it has been shown in [44–46] that comparable performance can be obtained without such complex manipulations, but with simple re-weighting the loss functions. Therefore, to encourage consensus between these gradients without modifying them, we propose aligning gradients between source and target domains during training. Specifically, we aim to maximize their cosine similarity, $\langle \boldsymbol{g}_{sh,S}, \boldsymbol{g}_{sh,T} \rangle$, If this goal is achieved, one can expect the shared prompt to capture useful features for classes regardless of domains. Indeed, $-\boldsymbol{g}_{sh,S}$ denotes the direction that moves the shared prompt towards low-loss region of source data, and similar for $-\boldsymbol{g}_{sh,T}$. Hence, when they point to the same direction, i.e., $\langle \boldsymbol{g}_{sh,S}, \boldsymbol{g}_{sh,T} \rangle > 0$, updating the shared prompt as in Eq. 6 can reduce loss of both domains, because the aggregated gradient $\boldsymbol{g}_{sh} = \lambda\boldsymbol{g}_{sh,S} + \boldsymbol{g}_{sh,T}$ will create acute angles with both $\boldsymbol{g}_{sh,S}$ and $\boldsymbol{g}_{sh,T}$. As a result, the shared prompt can learn knowledge that benefits both domains, which is its ultimate goal.

However, there remain two important questions when implementing this gradient alignment constrain: (i) How to incorporate the cosine similarity maximization term as a regularization in the framework described in Sec. 4.2?; and (ii) How to reduce training time and space when explicitly maximizing it, as it involves the computation of Hessian matrix w.r.t the shared prompt? Our method will address these two concerns.

Consider the following loss applied on target data with $||.||$ denoting $l_2$-norm of a vector:

$$\mathcal{L}_T^{\text{align}}(\boldsymbol{P}) := \mathcal{L}_T\left(\boldsymbol{P}_{sh} - \rho\frac{\boldsymbol{g}_{sh,S}}{\|\boldsymbol{g}_{sh,S}\|.\|\boldsymbol{g}_{sh,T}\|}, \boldsymbol{P}_T\right)$$
$$\approx \mathcal{L}_T(\boldsymbol{P}_{sh}, \boldsymbol{P}_T) - \rho\frac{(\boldsymbol{g}_{sh,S})^{\mathbb{T}}.\nabla_{\boldsymbol{P}_{sh}}\mathcal{L}_T(\boldsymbol{P}_{sh}, \boldsymbol{P}_T)}{\|\boldsymbol{g}_{sh,S}\|.\|\boldsymbol{g}_{sh,T}\|}$$
$$= \mathcal{L}_T(\boldsymbol{P}_{sh}, \boldsymbol{P}_T) - \rho\langle \boldsymbol{g}_{sh,S}, \boldsymbol{g}_{sh,T} \rangle, \tag{7}$$

where Eq. 7 is obtained by applying first-order Taylor expansion with $\rho$ is a small value, and $\mathbb{T}$ is the vector transpose. It can be seen that minimizing this loss also maximizes cosine similarity between gradients of the two domains. In order to achieve this, let denote $\boldsymbol{a} = \frac{\boldsymbol{g}_{sh,S}}{\|\boldsymbol{g}_{sh,S}\|.\|\boldsymbol{g}_{sh,T}\|}$, and consider the loss's gradient w.r.t $\boldsymbol{P}_{sh}$:

$$\boldsymbol{g}_{sh,T}^{\text{align}} := \nabla_{\boldsymbol{P}_{sh}}\mathcal{L}_T(\boldsymbol{P}_{sh} - \rho\boldsymbol{a}, \boldsymbol{P}_T)$$
$$= \frac{d(\boldsymbol{P}_{sh} - \rho\boldsymbol{a})}{d(\boldsymbol{P}_{sh})}\nabla_{\boldsymbol{P}_{sh}}\mathcal{L}_T(\boldsymbol{P}_{sh}, \boldsymbol{P}_T)\Big|_{\boldsymbol{P}_{sh}=\boldsymbol{P}_{sh}-\rho\boldsymbol{a}}$$
$$\approx \nabla_{\boldsymbol{P}_{sh}}\mathcal{L}_T(\boldsymbol{P}_{sh}, \boldsymbol{P}_T)|_{\boldsymbol{P}_{sh}=\boldsymbol{P}_{sh}-\rho\boldsymbol{a}}. \tag{8}$$

In the approximation of Eq. 8, we avoid the Hessian computation by dropping the derivative of $\boldsymbol{a}$ w.r.t $\boldsymbol{P}_{sh}$. Now we can practically apply deep learning optimizers, such as SGD, to minimize $\mathcal{L}_T^{\text{align}}(\boldsymbol{P})$. Specifically, we first compute gradients of the source and target losses w.r.t the shared prompt to get vector $\boldsymbol{a}$, then move the current shared prompt to the new stage: $\boldsymbol{P}_{sh} = \boldsymbol{P}_{sh} - \rho\boldsymbol{a}$. Finally, at this new stage, re-compute the loss on target data then calculate the new gradient.

In a similar way, we can derive $\mathcal{L}_S^{\text{align}}(\boldsymbol{P})$ on source data and then compute its new gradient w.r.t the shared prompt, i.e. $\boldsymbol{g}_{sh,S}^{\text{align}}$. Given these two new gradients, we can combine them to get the final update direction of the shared prompt, which will navigate it to common low-valued regions in the loss landscapes of both domains.

$$\boldsymbol{b} = \frac{\boldsymbol{g}_{sh,T}}{\|\boldsymbol{g}_{sh,S}\|\cdot\|\boldsymbol{g}_{sh,T}\|}, \boldsymbol{g}_{sh,S}^{\text{align}} \approx \nabla_{\boldsymbol{P}_{sh}}\mathcal{L}_S(\boldsymbol{P}_{sh},\boldsymbol{P}_S)|_{\boldsymbol{P}_{sh}=\boldsymbol{P}_{sh}-\rho\boldsymbol{b}},$$

$$\boldsymbol{g}_{sh}^{\text{align}} = \boldsymbol{g}_{sh,T}^{\text{align}} + \lambda\boldsymbol{g}_{sh,S}^{\text{align}}.$$

### 4.4 Prompt gradient-norm penalization for UDA

So far, we have proposed casting each domain loss as an objective in a multiple-objective optimization framework, and have suggested maximizing congruence between gradients of these objectives to reduce their inherent conflict. However, the domain loss is in the empirical form, which has been shown to be easily stuck in sharp minima and thus limiting generalization ability [81, 82], especially under distribution shifts [83]. Therefore, we argue that explicit control over the generalization of these prompts can be beneficial. Moreover, inspired by the finding in [59] that gradient norm penalization can help model favor flat minima, and by the effectiveness of such minima in the context of multi-task learning [81], we propose minimizing prompt gradient norm of each objective to enhance prompt generalization.

By following the same analysis as in Eq. 7, we can seamlessly fuse the gradient norm penalty term with the cosine similarity maximization with the loss below:

$$\mathcal{L}_T^{\text{PGA}}(\boldsymbol{P}) := \mathcal{L}_T\left(\boldsymbol{P}_{sh} - \rho_{ga}\frac{\boldsymbol{g}_{sh,S}}{\|\boldsymbol{g}_{sh,S}\|\cdot\|\boldsymbol{g}_{sh,T}\|} + \rho_{gn}\frac{\boldsymbol{g}_{sh,T}}{\|\boldsymbol{g}_{sh,T}\|}, \boldsymbol{P}_T + \rho_{gn}\frac{\boldsymbol{g}_T}{\|\boldsymbol{g}_T\|}\right)$$

$$\approx \mathcal{L}_T(\boldsymbol{P}_{sh},\boldsymbol{P}_T) - \rho_{ga}\frac{(\boldsymbol{g}_{sh,S})^{\mathbb{T}}.\nabla_{\boldsymbol{P}_{sh}}\mathcal{L}_T(\boldsymbol{P}_{sh},\boldsymbol{P}_T)}{\|\boldsymbol{g}_{sh,S}\|\cdot\|\boldsymbol{g}_{sh,T}\|} + \rho_{gn}(\|\boldsymbol{g}_{sh,T}\| + \|\boldsymbol{g}_T\|)$$

$$= \mathcal{L}_T(\boldsymbol{P}_{sh},\boldsymbol{P}_T) - \rho_{ga}\langle\boldsymbol{g}_{sh,S},\boldsymbol{g}_{sh,T}\rangle + \rho_{gn}(\|\boldsymbol{g}_{sh,T}\| + \|\boldsymbol{g}_T\|),$$

where $\boldsymbol{g}_T$ is the gradient of the target loss w.r.t target-specific $\boldsymbol{P}_T$, and $gn$ stands for gradient norm.

We then follow the derivation of Eq. 8 to come up with a practical approximation of the gradient of $\mathcal{L}_T^{\text{PGA}}(\boldsymbol{P})$

$$\boldsymbol{g}_{sh,T}^{\text{PGA}}, \boldsymbol{g}_T^{\text{PGA}} := \nabla_{\boldsymbol{P}}\mathcal{L}_T^{\text{PGA}}(\boldsymbol{P})$$

$$\approx \nabla_{\boldsymbol{P}}\mathcal{L}_T(\boldsymbol{P}_{sh},\boldsymbol{P}_T)|_{\boldsymbol{P}_{sh}=\boldsymbol{P}_{sh}-\rho_{ga}\boldsymbol{a}+\rho_{gn}\frac{\boldsymbol{g}_{sh,T}}{\|\boldsymbol{g}_{sh,T}\|},\boldsymbol{P}_T=\boldsymbol{P}_T+\rho_{gn}\frac{\boldsymbol{g}_T}{\|\boldsymbol{g}_T\|}}.$$

Similarly, we obtain the gradient of the source objective

$$\boldsymbol{g}_{sh,S}^{\text{PGA}}, \boldsymbol{g}_S^{\text{PGA}} \approx \nabla_{\boldsymbol{P}}\mathcal{L}_S(\boldsymbol{P}_{sh},\boldsymbol{P}_S)|_{\boldsymbol{P}_{sh}=\boldsymbol{P}_{sh}-\rho_{ga}\boldsymbol{b}+\rho_{gn}\frac{\boldsymbol{g}_{sh,S}}{\|\boldsymbol{g}_{sh,S}\|},\boldsymbol{P}_S=\boldsymbol{P}_S+\rho_{gn}\frac{\boldsymbol{g}_S}{\|\boldsymbol{g}_S\|}}.$$

Following the same update rules in Eq. 5 and Eq. 6, the prompts can be learnt to achieve both of our two goals: inter-domain gradient alignment and flat minima enforcement, which can lead to improved performance for UDA. We will recap this with a generalization bound in the next part, and provide details for the final loss function in Appendix B.

### 4.5 Theoretical Analysis of PGA

We informally present an information-theoretic bound to explain why PGA works. Refer to Appendix A for the formal version. For simplicity, we will consider the single-source UDA setting and abuse $N$ as the number of source samples. Let $\mathcal{Z}, \mathcal{P}$ be the input-label space and prompt space (or hypothesis

space), respectively. Assume the loss function $\ell : \mathcal{P} \times \mathcal{Z} \to \mathbb{R}_0^+$ is R-subgaussian [*] Denote $\mu, \mu'$ as the two underlying distributions from which the source and target data is sampled, and $KL(.||.)$ as the KL-divergence. The generalization error [*] is defined as the difference between the target population loss and the source empirical loss

$$Err := \mathbb{E}_{\boldsymbol{P}, D_S, D_T}[R_{\mu'}(\boldsymbol{P}) - R_{D_S}(\boldsymbol{P})] = \mathbb{E}_{\boldsymbol{P}, D_S, D_T}[\mathbb{E}_{\boldsymbol{Z}' \sim \mu'}[\ell(\boldsymbol{P}, \boldsymbol{Z}')] - \frac{1}{N}\sum_{i=1}^{N}\ell(\boldsymbol{P}, \boldsymbol{Z}_i)].$$

**Theorem 4.1.** *Under the assumption R-subgaussianity, the generalization error can be upper-bounded by:*

$$|Err| \leq \sqrt{\frac{4R^2}{N}\sum_{t=1}^{\mathcal{T}}\tilde{\eta}_t^2\mathbb{E}_{\boldsymbol{P}_{t-1}, D_S, D_T}[\|\boldsymbol{g}_t^{src}\|^2 + \|\boldsymbol{g}_t^{tgt}\|^2 + \|\boldsymbol{g}_t^{src} - \boldsymbol{g}_t^{tgt}\|^2]} + \sqrt{2R^2KL(\mu||\mu')},$$

*where $\mathcal{T}$ is the total number of training iterations, $\tilde{\eta}_t$ is the learning rate at iteration $t$ scaled by a scalar, $\boldsymbol{g}_t^{src} = \nabla_{\boldsymbol{P}}\mathcal{L}_S(\boldsymbol{P}_{t-1})$, $\boldsymbol{g}_t^{tgt} = \nabla_{\boldsymbol{P}}\mathcal{L}_T(\boldsymbol{P}_{t-1})$ are the gradients w.r.t $\boldsymbol{P}_{t-1}$ of source loss Eq. 2 and target loss Eq 3 where $\boldsymbol{P}_t$ is the prompt at iteration $t$.*

As our method tries to minimize source empirical loss, gradient norms and gradient mis-alignment, from the first term in the R.H.S of Eq. 4.1, its benefit to the performance on target domain can be justified. Furthermore, the second term shows that the generalization error can be reduced by bridging the gap between the two domain distributions, which is the core of many UDA methods, such as [70, 84]. However, as stated earlier, our work is orthogonal to this line of method as we do not explicitly attempt to close such gap. Hence, an interesting future development could be taking the second term into account. Refer to Appendix A.5 for more discussion about this bound.

# 5 Experiments

In this section, we evaluate the efficacy of our proposed method on different UDA benchmarks, following the same protocol of recent prompt-based UDA studies [25, 28]. Before that, we start with a simple multi-objective-optimization setup to derive insights into the effectiveness of our proposed method compared to conventional empirical risk minimization (ERM).

## 5.1 Illustrative example

Let $\mathbf{y} \in \{-1, 1\}$ be the true label, $\mathbf{e}$ be the environmental feature and $\epsilon$ be Gaussian noise, $\mathbf{x} \in \mathbb{R}^{300}$, and $\mathrm{p} \in (0, 1), C > 1$ be predefined scalar constants. The data-generating process is given by:

$$\mathbf{y} \sim \mathcal{U}\{-1, 1\}, \quad \mathbf{e} \sim \begin{cases} p_{\mathrm{p}}(\mathbf{e} = y \mid \mathbf{y} = y) = \mathrm{p} \\ p_{\mathrm{p}}(\mathbf{e} = -y \mid \mathbf{y} = y) = (1 - \mathrm{p}) \end{cases}, \quad \epsilon \sim \mathcal{N}\left(0, \mathbf{I}^{298}\right), \quad \mathbf{x} = [C * \mathbf{e}, \mathbf{y}, \epsilon]$$

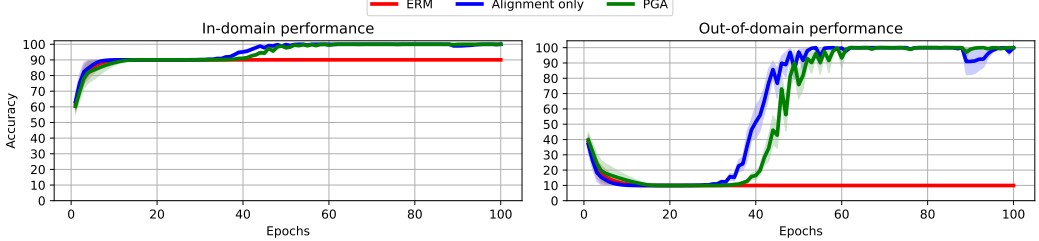

Figure 2: Performance of ERM and PGA on the in-domain data (validation set) and out-of-distribution data (test set). Average results and shaded standard errors are obtained from 10 random seeds.

The environmental feature $\mathbf{e}$ correlates with the true label $\mathbf{y}$ according to p. Similar to [55, 85], we set $\mathrm{p} = 0.9$ for the training and validation set (in-distribution) and $\mathrm{p} = 0.1$ for the test set

---

[*] A random variable $X$ is R-subgaussian if for any $\rho$, $\log \mathbb{E}\exp(\rho(X - \mathbb{E}X)) \leq \rho^2 R^2/2$.

[*] Refer to the Appendix to see why the expectation is taken over $\boldsymbol{P}, D_S, D_T$.

Table 2: Accuracy (%) on ImageCLEF and Office-Home. We use **bold** to denote the best method overall and underscore to denote the best method using source combined. Overall, PGA and MPGA consistently obtain the best results among source combined and multi-source scenarios, respectively.

| | ImageCLEF | | | | Office-Home | | | | |
| | $\rightarrow$ C | $\rightarrow$ I | $\rightarrow$ P | Avg | $\rightarrow$ Ar | $\rightarrow$ Cl | $\rightarrow$ Pr | $\rightarrow$ Rw | Avg |
|---|---|---|---|---|---|---|---|---|---|
| **Zero-Shot** | | | | | | | | | |
| CLIP [27] | 87.9 | 88.2 | 78.7 | 88.1 | 71.2 | 50.4 | 81.4 | 82.6 | 71.4 |
| **Source Combined** | | | | | | | | | |
| DAN [19] | 93.3 | 92.2 | 77.6 | 87.7 | 68.5 | 59.4 | 79.0 | 82.5 | 72.4 |
| DANN [18] | 93.7 | 91.8 | 77.9 | 87.8 | 68.4 | 59.1 | 79.5 | 82.7 | 72.4 |
| D-CORAL [69] | 93.6 | 91.7 | 77.1 | 87.5 | 68.1 | 58.6 | 79.5 | 82.7 | 72.2 |
| DAPL [25] | 96.0 | 89.2 | 76.0 | 87.1 | 72.8 | 51.9 | 82.6 | 83.7 | 72.8 |
| Simple Prompt [28] | 93.6 | 90.6 | 80.9 | 88.4 | 70.7 | 52.9 | 82.9 | 83.9 | 72.4 |
| **PGA** (Ours) | 96.8 | 95.7 | 84.6 | 92.4 | 75.2 | 59.7 | 86.2 | 86.2 | 76.8 |
| **Multi-Source** | | | | | | | | | |
| DCTN [88] | 95.7 | 90.3 | 75.0 | 87.0 | N.A. | N.A. | N.A. | N.A. | N.A. |
| MDDA [89] | N.A. | N.A. | N.A. | N.A. | 66.7 | 62.3 | 79.5 | 79.6 | 71.0 |
| SIMplDA [96] | 93.3 | 91.0 | 77.5 | 87.3 | 70.8 | 56.3 | 80.2 | 81.5 | 72.2 |
| MFSAN [62] | 95.4 | 93.6 | 79.1 | 89.4 | 72.1 | 62.0 | 80.3 | 81.8 | 74.1 |
| MPA [28] | 97.2 | 96.2 | 80.4 | 91.3 | 74.8 | 54.9 | 86.2 | 85.7 | 75.4 |
| **MPGA** (Ours) | **97.4** | **96.5** | **84.7** | **92.9** | **76.3** | **63.8** | **90.0** | **87.4** | **79.4** |

(out-of-distribution). Figure 2 presents the performance of three linear classifiers trained by ERM, our gradient alignment method only and PGA. In summary, while ERM learns non-predictive features and fails to generalize beyond in-distribution data, our gradient alignment algorithm can leverage the gradient information from multiple environments to learn the core feature that helps perform well on OOD data. Besides, incorporating the gradient norm penalty term further enhances stability and robustness at convergence.

## 5.2 Experimental setup

**Datasets**. We conduct experiments using three well-established UDA datasets of varying scales: ImageCLEF [17], Office-Home [86], and DomainNet [87], respectively. Detailed descriptions of these datasets are available in Appendix C.1.

**Metrics**. We evaluate our model by reporting the top-1 accuracy for each target domain and the average accuracy across all domains. To further validate the effectiveness of our proposed method, we conduct experiments in two distinct settings: a source-combined setting, where data from all source domains are merged, and a multi-source setting, which utilizes individual domain identifications. Additionally, we provide pair-wise single-source domain adaptation results for the Office-Home dataset.

**Baselines.** Regarding prompt-based baselines, we compare our method with MPA [28], DAPL [25], Simple Prompt [28], and Zero-shot CLIP [27]. To ensure a comprehensive evaluation, we also include comparisons with various non-prompt methods such as DCTN [88], MDDA [89], MFSAN [62], T-SVDNet [90] and PFSA [91] ... As we follow the same settings as in [28] and [25], results for baselines are taken from those studies for the consistency. Note that while DAPL [25], MPA [28] and our methods employ CoOp [92] with text-end soft-prompt, other methods finetune the transformer block [63] or both image and text-end soft-prompts [93] or the whole encoders [94, 95]. Since those methods typically fine-tune many more parameters, we thus do not include them in the experimental results for the sake of fair comparison.

## 5.3 Experimental results

Table 2 presents the results for the ImageCLEF and Office-Home datasets. Under the source-combined scenario, PGA significantly outperforms nearly all other baseline methods on both datasets, with the exception of its own multi-source variant, MPGA. For instance, PGA surpasses the second-

best source combined method by a notable $4\%$ in average accuracy and exceeds MPA by over $1\%$. Notably, in the Office-Home domain Clipart, while two prompt-based baselines, DAPL and Simple Prompt, lag behind their non-prompt counterparts, PGA still manages to achieve slightly better results than these non-prompt methods. In the multi-source setting, MPGA consistently delivers the highest performance across all domains, notably outperforming MPA, the state-of-the-art (SOTA) prompt-based method for multi-source UDA, by a substantial margin of $4\%$ on Office-Home.

Table 3: Accuracy (%) on DomainNet. We use **bold** to denote the best method overall and underscore to denote the best method using source combine.

| | DomainNet | | | | | | |
|---|---|---|---|---|---|---|---|
| | $\rightarrow$ **Clp** | $\rightarrow$ **Inf** | $\rightarrow$ **Pnt** | $\rightarrow$ **Qdr** | $\rightarrow$ **Rel** | $\rightarrow$ **Skt** | **Avg** |
| **Zero-Shot** | | | | | | | |
| CLIP [27] | 61.3 | 42.0 | 56.1 | 10.3 | 79.3 | 54.1 | 50.5 |
| **Source Combined** | | | | | | | |
| DANN [18] | 45.5 | 13.1 | 37.0 | 13.2 | 48.9 | 31.8 | 32.6 |
| MCD [97] | 54.3 | 22.1 | 45.7 | 7.6 | 58.4 | 43.5 | 38.5 |
| DAPL [25] | 62.4 | 43.8 | 59.3 | 10.6 | 81.5 | 54.6 | 52.0 |
| Simple Prompt [28] | 63.1 | 41.2 | 57.7 | 10.0 | 75.8 | 55.8 | 50.6 |
| **PGA** (Ours) | 66.3 | 49.2 | 63.3 | 11.1 | 81.8 | 60.6 | 55.4 |
| **Multi-Source** | | | | | | | |
| M³SDA-$\beta$ [98] | 58.6 | 26.0 | 52.3 | 6.3 | 62.7 | 49.5 | 42.6 |
| SImpAl101 [96] | 66.4 | 26.5 | 56.6 | **18.9** | 68.0 | 55.5 | 48.6 |
| LtC-MSDA [99] | 63.1 | 28.7 | 56.1 | 16.3 | 66.1 | 53.8 | 47.4 |
| T-SVDNet [90] | 66.1 | 25.0 | 54.3 | 16.5 | 65.4 | 54.6 | 47.0 |
| PFSA [91] | 64.5 | 29.2 | 57.6 | 17.2 | 67.2 | 55.1 | 48.5 |
| PTMDA [100] | 66.0 | 28.5 | 58.4 | 13.0 | 63.0 | 54.1 | 47.2 |
| MPA [28] | 65.2 | 47.3 | 62.0 | 10.2 | 82.0 | 57.9 | 54.1 |
| **MPGA** (Ours) | **67.9** | **50.5** | **63.8** | 11.6 | **82.2** | **61.0** | **56.2** |

On DomainNet, as Table 3 presents, our method still obtains superior average accuracy under both source combined and multi-source, higher than the runner-up by 3.4% and 2.1%, respectively. Overall, in the domain where CLIP brings significant results compared with non-prompt baselines, our method leads to better performance, except for the difficult QuickDraw domain, as remarked by a relatively low zero-shot accuracy for CLIP-based methods, where it seems that prompt learning fails to beat non-prompt counterparts. Even though, both PGA and MPGA still outperform other prompt-based counterparts while fine-tuning fewer parameters (e.g. 500k versus 2M of MPA).

In addition, we also demonstrate our method's effectiveness under 12 pair-wise source-target settings on Office-Home in Table 4. Again, PGA acquires the highest average score and consistently beats DAPL under 12 settings while using the same parameter-efficient-finetuning method [92].

Table 4: Accuracy (%) on Office-Home[101] for unsupervised domain adaptation (ResNet-50[102]). The best accuracy is indicated in **bold**.

| Method | Ar→Cl | Ar→Pr | Ar→Rw | Cl→Ar | Cl→Pr | Cl→Rw | Pr→Ar | Pr→Cl | Pr→Rw | Rw→Ar | Rw→Cl | Rw→Pr | Avg |
|---|---|---|---|---|---|---|---|---|---|---|---|---|---|
| ResNet-50[102] | 34.9 | 50.0 | 58.0 | 37.4 | 41.9 | 46.2 | 38.5 | 31.2 | 60.4 | 53.9 | 41.2 | 59.9 | 46.1 |
| DANN [19] | 45.6 | 59.3 | 70.1 | 47.0 | 58.5 | 60.9 | 46.1 | 43.7 | 68.5 | 63.2 | 51.8 | 76.8 | 57.6 |
| JAN [17] | 45.9 | 61.2 | 68.9 | 50.4 | 59.7 | 61.0 | 45.8 | 43.4 | 70.3 | 63.9 | 52.4 | 76.8 | 58.3 |
| CDAN+E [71] | 50.7 | 70.6 | 76.0 | 57.6 | 70.0 | 70.0 | 57.4 | 50.9 | 77.3 | 70.9 | 56.7 | 81.6 | 65.8 |
| BSP+CDAN [103] | 52.0 | 68.6 | 76.1 | 58.0 | 70.3 | 70.2 | 58.6 | 50.2 | 77.6 | 72.2 | 59.3 | 81.9 | 66.3 |
| SymNets [104] | 47.7 | 72.9 | 78.5 | 64.2 | 71.3 | 74.2 | 63.6 | 47.6 | 79.4 | 73.8 | 50.8 | 82.6 | 67.2 |
| ETD [105] | 51.3 | 71.9 | 85.7 | 57.6 | 69.2 | 73.7 | 57.8 | 51.2 | 79.3 | 70.2 | 57.5 | 82.1 | 67.3 |
| BNM [106] | 52.3 | 73.9 | 80.0 | 63.3 | 72.9 | 74.9 | 61.7 | 49.5 | 79.7 | 70.5 | 53.6 | 82.2 | 67.9 |
| GSDA [107] | **61.3** | 76.1 | 79.4 | 65.4 | 73.3 | 74.3 | 65.0 | 53.2 | 80.0 | 72.2 | **60.6** | 83.1 | 70.3 |
| GVB-GD [108] | 57.0 | 74.7 | 79.8 | 64.6 | 74.1 | 74.6 | 65.2 | 55.1 | 81.0 | 74.6 | 59.7 | 84.3 | 70.4 |
| RSDA-MSTN [109] | 53.2 | 77.7 | 81.3 | 66.4 | 74.0 | 76.5 | 67.9 | 53.0 | 82.0 | 75.8 | 57.8 | 85.4 | 70.9 |
| SPL [110] | 54.5 | 77.8 | 81.9 | 65.1 | 78.0 | 81.1 | 66.0 | 53.1 | 82.8 | 69.9 | 55.3 | **86.0** | 71.0 |
| SRDC [26] | 52.3 | 76.3 | 81.0 | 69.5 | 76.2 | 78.0 | 68.7 | 53.8 | 81.7 | **76.3** | 57.1 | 85.0 | 71.3 |
| DisClusterDA [111] | 58.8 | 77.0 | 80.8 | 67.0 | 74.6 | 77.1 | 65.9 | **56.3** | 81.4 | 74.2 | 60.5 | 83.6 | 71.4 |
| CLIP [27] | 51.6 | 81.9 | 82.6 | 71.9 | 81.9 | 82.6 | 71.9 | 51.6 | 82.6 | 71.9 | 51.6 | 81.9 | 72.0 |
| DAPL [25] | 54.1 | 84.3 | 84.8 | 74.4 | 83.7 | 85.0 | 74.5 | 54.6 | 84.8 | 75.2 | 54.7 | 83.8 | 74.5 |
| **PGA** (Ours) | 56.1 | **85.5** | **86.0** | **75.5** | **85.2** | **85.8** | **75.2** | 55.7 | **86.1** | 75.4 | 56.7 | 85.8 | **75.8** |

## 5.4 Ablation study

From Table 5, we can see that (i) learning prompts using solely the target loss, the accuracy across all settings already surpasses that of Zero-shot CLIP. This confirms the reliability of pseudo labels generated by CLIP. (ii) When adding source loss and grad-norm penalization, the results improve slightly. (iii) Importantly, adding gradient alignment, the scores increase more clearly. These observations verify each of our contributions.

| $L_T$ | $L_S$ | GN | GA | $\rightarrow$ C | $\rightarrow$ I | $\rightarrow$ P | Avg |
|---|---|---|---|---|---|---|---|
| $\times$ | $\times$ | $\times$ | $\times$ | 87.9 | 88.2 | 78.7 | 88.1 |
| $\checkmark$ | $\times$ | $\times$ | $\times$ | 92.9 | 94.3 | 83.2 | 90.1 |
| $\checkmark$ | $\checkmark$ | $\times$ | $\times$ | 93.3 | 95.0 | 83.3 | 90.6 |
| $\checkmark$ | $\checkmark$ | $\checkmark$ | $\times$ | 94.3 | 95.3 | 83.2 | 90.9 |
| $\checkmark$ | $\checkmark$ | $\checkmark$ | $\checkmark$ | 96.8 | 95.7 | 84.6 | 92.4 |

Table 5: Ablation studies on various modules of PGA on the ImageCLEF. Each of the proposed components shows its effectiveness while combining them helps obtain the best performance.

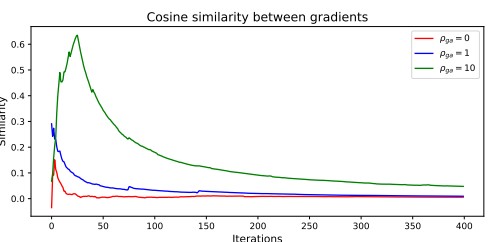

Figure 3: Evolution of the gradient similarity during training.

Furthermore, to show that gradient alignment indeed increases consensus between gradients, we plot cosine similarity along the training process with three different values of $\rho_{ga}$ in Figure 3. First, during early training stages, there seems to be less agreement between gradients when no alignment is enforced, c.f. $\rho_{ga} = 0$. When $\rho_{ga} > 0$, we can see the similarity increase. Noticeably, there exists a point where similarity starts plummeting. This is reasonable when the model starts to converge to a Pareto solution where source and target gradients cancel each other. This is depicted more clearly in Figure 4 in the appendix where the closer the model is to the Pareto front, the more conflict the gradients are.

## 6 Conclusion

In this work, we have proposed a framework for UDA inspired by Multi-objective optimization thanks to the generalizability of CLIP and the lightweight nature of prompt learning. We have then devised a practical method to align per-objective gradients, which aims to encourage inherent consensus between objectives. We have further fused gradient norm penalization into the method to enhance prompt generalization. Finally, a UDA generalization bound is presented to justify the benefits of our method.

**Acknowledgements**: Trung Le was supported by ARC DP23 grant DP230101176 and by the Air Force Office of Scientific Research under award number FA2386-23-1-4044.

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

# Supplement to "Enhancing Domain Adaptation through Prompt Gradient Alignment"

Due to space constraints, some details were omitted from the main paper. We therefore include additional theoretical developments (section A), the detailed algorithm description (section B) and experimental results (section C) in this appendix.

## A  UDA generalization bound

Here, we provide an information-theoretic generalization bound for UDA, which can be reduced by our gradient alignment and gradient norm penalization. For simplicity, we will consider the single-source UDA setting.

To begin, we first define some additional notations: let $\mathcal{X}, \mathcal{Y}, \mathcal{P}$ be the input space, output space, and prompt space (or hypothesis space), respectively. Denote the input-label space as $\mathcal{Z} = \mathcal{X} \times \mathcal{Y}$, and the loss function as $\ell : \mathcal{P} \times \mathcal{Z} \to \mathbb{R}_0^+$ (Cross entropy in our case). Finally, denote $\mu, \mu'$ as the two underlying distributions from which the source and target domains are sampled. The training data for source domain $D_S = \{\boldsymbol{Z}_i\}_{i=1}^N$ is drawn i.i.d from $\mu^{\otimes N}$, and that for target domain $D_T = \{\boldsymbol{X}_j'\}_{j=1}^M$ is from $\mu_X'^{\otimes M}$.

For each prompt parameter, the population risk in the target domain is defined as

$$R_{\mu'}(\boldsymbol{P}) := \mathbb{E}_{\boldsymbol{Z}' \sim \mu'}[\ell(\boldsymbol{P}, \boldsymbol{Z}')]. \tag{9}$$

This risk is the ultimate goal that a UDA algorithm aims to minimize. However, since $\mu'$ is unknown, and only a finite number of training data is given, we define the empirical risk in the source domain as

$$R_{D_S}(\boldsymbol{P}) := \frac{1}{N} \sum_{i=1}^N \ell(\boldsymbol{P}, \boldsymbol{Z}_i). \tag{10}$$

In the information-theoretic analysis framework, model parameter, $\boldsymbol{P} \in \mathcal{P}$ in our case, is a random variable that is outputted from a learning algorithm $\mathcal{A}$ characterized by some conditional distribution $P_{\boldsymbol{P}|D_S, D_T}$. Then the generalization error, measuring how close these two risks can be, has the form

$$Err := \mathbb{E}_{\boldsymbol{P}, D_S, D_T}[R_{\mu'}(\boldsymbol{P}) - R_{D_S}(\boldsymbol{P})], \tag{11}$$

where the expectation is taken over $\boldsymbol{P}, D_S, D_T \sim P_{\boldsymbol{P}|D_S, D_T}, \mu^{\otimes N}, \mu_X'^{\otimes M}$.

To derive the bound, we need the following assumption on the loss function, which is commonly adopted in many information-theoretic bounds such as those in [112, 113]:

**Assumption A.1.** (Subgaussianity). $\ell(\boldsymbol{P}; \boldsymbol{Z}')$ is R-subgaussian $^*$ under $P_{\boldsymbol{P}, \boldsymbol{Z}'|\boldsymbol{X}_j' = \boldsymbol{x}_j'}, \forall \boldsymbol{x}_j' \in \mathcal{X}$, for any $\boldsymbol{P} \in \mathcal{P}$.

We also present the definitions of Mutual Information, Disintegrated Mutual Information, and Conditional Mutual Information:

**Definition A.2.** (Mutual Information). $I(X; Y) = \mathrm{KL}(P_{X,Y} || P_X \otimes P_Y)$, where KL is the KL-divergence and $\otimes$ denote the product of two marginal distributions.

**Definition A.3.** (Disintegrated Mutual Information). The disintegrated mutual information between two random variables $X$ and $Y$ given a realization of a variable $Z = z$ is

$$I^{Z=z}(X; Y) = \mathrm{KL}(P_{X,Y|Z=z} || P_{X|Z=z} \otimes P_{Y|Z=z})$$

**Definition A.4.** (Conditional Mutual Information). $I(X, Y|Z) = \mathbb{E}_Z I^Z(X; Y)$.

---

$^*$A random variable $X$ is R-subgaussian if for any $\rho, \log \mathbb{E} \exp(\rho(X - \mathbb{E}X)) \leq \rho^2 R^2 / 2$.

**Theorem A.5.** *Under assumption A.1, the generalization error can be upper-bounded by*

$$|Err| \leq \sqrt{\frac{4R^2}{N} \sum_{t=1}^{\mathcal{T}} \frac{\eta_t^2}{\sigma_t^2} \mathbb{E}_{\boldsymbol{P}_{t-1}, D_S, D_T}[\|\boldsymbol{g}_t^{src}\|^2 + \|\boldsymbol{g}_t^{tgt}\|^2 + \|\boldsymbol{g}_t^{src} - \boldsymbol{g}_t^{tgt}\|^2]} + \sqrt{2R^2 KL(\mu||\mu')},$$

(12)

*where $\mathcal{T}$ is the total number of training iterations, $\eta_t$ is the learning rate at iteration $t$, $\boldsymbol{P}_t$ is the prompt at iteration $t$, $\boldsymbol{g}_t^{src} = \nabla_{\boldsymbol{P}} \mathcal{L}_{src}(\boldsymbol{P}_{t-1})$, $\boldsymbol{g}_t^{tgt} = \nabla_{\boldsymbol{P}} \mathcal{L}_{tgt}(\boldsymbol{P}_{t-1})$ are the gradients w.r.t $\boldsymbol{P}_{t-1}$ of source loss Eq.2 and target loss Eq.3, and $\sigma_t$ is the standard deviation of the isotropic Gaussian noise added to the update of $\boldsymbol{P}_t$.*

**Remark A.6.** *For the purpose of simplicity, here we consider a 'noisy' update version of prompts: $\boldsymbol{P}_t = \boldsymbol{P}_{t-1} - \eta_t \boldsymbol{g} + N_t, N_t \sim \mathcal{N}(\boldsymbol{0}, \sigma_t^2 \boldsymbol{I})$. However, note that the bound still holds for the conventional SGD update, i.e., no added noise, by following techniques in [113].*

**Remark A.7.** *Our methods align gradients of shared-prompt, but here we can omit its subscript in the inter-domain gradient matching term, $\|\boldsymbol{g}_t^{src} - \boldsymbol{g}_t^{tgt}\|^2$, by noting that $\boldsymbol{g}_t^{src} = [\boldsymbol{g}_t^{sh,src}, \boldsymbol{g}_t^S, \boldsymbol{0}]$ and $\boldsymbol{g}_t^{tgt} = [\boldsymbol{g}_t^{sh,tgt}, \boldsymbol{0}, \boldsymbol{g}_t^T]$. Indeed, $\|\boldsymbol{g}_t^{src} - \boldsymbol{g}_t^{tgt}\|^2 = \|\boldsymbol{g}_t^{src}\|^2 + \|\boldsymbol{g}_t^{tgt}\|^2 - 2(\boldsymbol{g}_t^{sh,src})^{\mathbb{T}} \boldsymbol{g}_t^{sh,tgt}$, where $\mathbb{T}$ denotes the vector transpose. In addition, this bound suggests maximizing the dot product between gradients; however, to stabilize training, we aim to maximize the cosine similarity instead.*

This theorem suggests that penalizing gradient norm and matching gradients across domains can improve generalization on the target domain, i.e., the first term in the R.H.S of A.5 is minimized. Note that minimizing gradient norm has been widely used in [59, 81, 114] to control the sharpness of the loss landscape, which is strongly related to the generalization capability of the model. In this work, we can empirically and theoretically verify the effectiveness of this technique in the gradient space of prompt, consistent with results in previous works [115, 116].

Regarding the second term, we do not aim for a method that can explicitly reduce the gap between source and target distributions, because we do not want to remove any domain-specific features that may be helpful for prediction. Instead, we want to capture domain-agnostic features in the shared prompt, and specific features in the domain-specific ones so that at inference, a more meaningful representation can be obtained by using these prompts. Hence, one possible direction for future work is to design and learn prompts such that domain distribution alignment can also be achieved.

Finally, this bound can grow as the number of training iterations increases unless gradient norms and the difference between source and target gradients are extremely small at final iterations. Future work could be overcoming this limitation by considering other bounds, such as ones suggested in [117].

*Proof.* Our bound is inspired from the bound in Theorem 5.1 in [112], which is restated as the following lemma

**Lemma A.8.** *Under assumption A.1, the generalization error can be upper-bounded by*

$$|Err| \leq \frac{1}{NM} \sum_{j=1}^{M} \sum_{i=1}^{N} \mathbb{E}_{\boldsymbol{X}_j'} \sqrt{2R^2 I^{\boldsymbol{X}_j'}(\boldsymbol{P}; \boldsymbol{Z}_i)} + \sqrt{2R^2 KL(\mu||\mu')}$$

(13)

$$\leq \sqrt{\frac{2R^2}{N} I(\boldsymbol{P}; D_S|D_T)} + \sqrt{2R^2 KL(\mu||\mu')}$$

(14)

Now consider the 'noisy' update of the prompt as presented in Eqs. 6 and 5:

$$\boldsymbol{P}_t = \boldsymbol{P}_{t-1} - \eta_t(\nabla_{\boldsymbol{P}} \mathcal{L}_{src}(\boldsymbol{P}_{t-1}) + \nabla_{\boldsymbol{P}} \mathcal{L}_{tgt}(\boldsymbol{P}_{t-1})) + N_t$$

(15)

$$:= \boldsymbol{P}_{t-1} - \eta_t \boldsymbol{g}_t^{src} - \eta_t \boldsymbol{g}_t^{tgt} + N_t.$$

(16)

Assume that we obtain the final prompts after $\mathcal{T}$ iterations, then following the chain rule of mutual information and data processing inequality, we have

$$I(\boldsymbol{P}_{\mathcal{T}}; D_S|D_T) = I(\boldsymbol{P}_{\mathcal{T}-1} - \eta_{\mathcal{T}}\boldsymbol{g}_{\mathcal{T}}^{src} - \eta_{\mathcal{T}}\boldsymbol{g}_{\mathcal{T}}^{tgt} + N_{\mathcal{T}}; D_S|D_T) \tag{17}$$

$$\leq I\left(\boldsymbol{P}_{\mathcal{T}-1}, -\eta_{\mathcal{T}}\boldsymbol{g}_{\mathcal{T}}^{src} - \eta_{\mathcal{T}}\boldsymbol{g}_{\mathcal{T}}^{tgt} + N_{\mathcal{T}}; D_S|D_T\right) \tag{18}$$

$$= I(\boldsymbol{P}_{\mathcal{T}-1}; D_S|D_T) + I(-\eta_{\mathcal{T}}\boldsymbol{g}_{\mathcal{T}}^{src} - \eta_{\mathcal{T}}\boldsymbol{g}_{\mathcal{T}}^{tgt} + N_{\mathcal{T}}; D_S|D_T, \boldsymbol{P}_{\mathcal{T}-1}) \tag{19}$$

$$\vdots \tag{20}$$

$$\leq \sum_{t=1}^{\mathcal{T}} I(-\eta_t\boldsymbol{g}_t^{src} - \eta_t\boldsymbol{g}_t^{tgt} + N_t; D_S|D_T, \boldsymbol{P}_{t-1}) \tag{21}$$

$$= \sum_{t=1}^{\mathcal{T}} I(-\boldsymbol{g}_t^{src} - \boldsymbol{g}_t^{tgt} + N_t/\eta_t; D_S|D_T, \boldsymbol{P}_{t-1}) \tag{22}$$

$$\leq \sum_{t=1}^{\mathcal{T}} I\left(-\boldsymbol{g}_t^{src} + \frac{N_t}{2\eta_t}, -\boldsymbol{g}_t^{tgt} + \frac{N_t}{2\eta_t}; D_S|D_T, \boldsymbol{P}_{t-1}\right) \tag{23}$$

$$= \sum_{t=1}^{\mathcal{T}} I\left(-\boldsymbol{g}_t^{src} + \frac{N_t}{2\eta_t}; D_S|D_T, \boldsymbol{P}_{t-1}\right)$$

$$+ I\left(-\boldsymbol{g}_t^{tgt} + \frac{N_t}{2\eta_t}; D_S|D_T, \boldsymbol{P}_{t-1}, -\boldsymbol{g}_t^{src} + \frac{N_t}{2\eta_t}\right) \tag{24}$$

Eq. 21 is due to the assumption of independence of $\boldsymbol{P}_0$ w.r.t $D_S$ and $D_T$, and Eq. 22 is because mutual information is scale-invariant.

Consider the first term in Eq. 24, for all $t$, inspired by the proof of Lemma 3 in [118], we have

$$I\left(-\boldsymbol{g}_t^{src} + \frac{N_t}{2\eta_t}; D_S|D_T, \boldsymbol{P}_{t-1}\right)$$

$$= \mathbb{E}_{D_S,D_T,\boldsymbol{P}_{t-1}}\left[\mathrm{KL}\left(P_{-\boldsymbol{g}_t^{src}+\frac{N_t}{2\eta_t}|D_T,\boldsymbol{P}_{t-1},D_S}||P_{-\boldsymbol{g}_t^{src}+\frac{N_t}{2\eta_t}|D_T,\boldsymbol{P}_{t-1}}\right)\right] \tag{25}$$

$$= \mathbb{E}_{D_S,D_T,\boldsymbol{P}_{t-1}}\left[\mathrm{KL}\left(P_{-\boldsymbol{g}_t^{src}+\frac{N_t}{2\eta_t}|D_T,\boldsymbol{P}_{t-1},D_S}||P_{\tilde{\boldsymbol{G}}_t|D_T,\boldsymbol{P}_{t-1}}\right) - \mathrm{KL}\left(P_{-\boldsymbol{g}_t^{src}+\frac{N_t}{2\eta_t}|D_T,\boldsymbol{P}_{t-1}}||P_{\tilde{\boldsymbol{G}}_t|D_T,\boldsymbol{P}_{t-1}}\right)\right] \tag{26}$$

$$\leq \mathbb{E}_{D_S,D_T,\boldsymbol{P}_{t-1}}\left[\mathrm{KL}\left(P_{-\boldsymbol{g}_t^{src}+\frac{N_t}{2\eta_t}|D_T,\boldsymbol{P}_{t-1},D_S}||P_{\tilde{\boldsymbol{G}}_t|D_T,\boldsymbol{P}_{t-1}}\right)\right], \tag{27}$$

where $P_{\tilde{\boldsymbol{G}}_t|D_T,\boldsymbol{P}_{t-1}}$ is some random distribution, every choice of which results in a upper bound for the MI, and the equality holds when $P_{\tilde{\boldsymbol{G}}_t|D_T,\boldsymbol{P}_{t-1}} = P_{-\boldsymbol{g}_t^{src}+\frac{N_t}{2\eta_t}|D_T,\boldsymbol{P}_{t-1}}$.

Therefore, if we choose $P_{\tilde{\boldsymbol{G}}_t|D_T,\boldsymbol{P}_{t-1}} = \mathcal{N}(0, \frac{\sigma_t^2}{4\eta_t^2}\boldsymbol{I})$, the R.H.S of Eq. 27 will be upper-bounded by $\frac{2\eta_t^2}{\sigma_t^2}\mathbb{E}_{D_S,D_T,\boldsymbol{P}_{t-1}}\left[||\boldsymbol{g}_t^{src}||^2\right]$, which is derived from the KL-divergence between two Gaussian distributions.

Similarly for the second term in Eq. 24, choosing $P_{\tilde{\boldsymbol{G}}_t|D_T,\boldsymbol{P}_{t-1},-\boldsymbol{g}_t^{src}+\frac{N_t}{2\eta_t}} = \mathcal{N}(0, \frac{\sigma_t^2}{4\eta_t^2}\boldsymbol{I})$ gives us the upper bound $\frac{2\eta_t^2}{\sigma_t^2}\mathbb{E}_{D_S,D_T,\boldsymbol{P}_{t-1}}\left[||\boldsymbol{g}_t^{tgt}||^2\right]$. Furthermore, letting $P_{\tilde{\boldsymbol{G}}_t|D_T,\boldsymbol{P}_{t-1},-\boldsymbol{g}_t^{src}+\frac{N_t}{2\eta_t}} = P_{-\boldsymbol{g}_t^{src}+\frac{N_t}{2\eta_t}}$, which is also a Gaussian distribution due to the effect of the added noise, we reach the gradient matching term, $\frac{2\eta_t^2}{\sigma_t^2}\mathbb{E}_{D_S,D_T,\boldsymbol{P}_{t-1}}\left[||\boldsymbol{g}_t^{src} - \boldsymbol{g}_t^{tgt}||^2\right]$.

Note that in Eq. 15, we suppose the source-weight term $\lambda = 1$ to simplify proof. However, if one wishes to keep the impact of $\lambda$, they can change $\boldsymbol{g}_t^{src} = [\lambda\boldsymbol{g}_t^{sh,src}, \boldsymbol{g}_t^S, \boldsymbol{0}]$. In this case, the terms under the expectation in the bound will become: $(\lambda^2-1)||\boldsymbol{g}_t^{sh,src}||^2 + ||\boldsymbol{g}_t^{src}||^2 + ||\boldsymbol{g}_t^{tgt}||^2 + ||\boldsymbol{g}_t^{tgt} - \lambda\boldsymbol{g}_t^{src}||^2$.

Combining everything together, the proof is done. $\qquad\square$

# B Algorithm

## B.1 Final objectives

As we cast UDA as a MOO problem, the ideal final objectives, in the case of single-source UDA, would be

$$[\mathcal{L}_S^{\text{PGA}}(\boldsymbol{P}), \mathcal{L}_T^{\text{PGA}}(\boldsymbol{P})],$$

where

$$\mathcal{L}_T^{\text{PGA}}(\boldsymbol{P}) := \mathcal{L}_T(\boldsymbol{P}_{sh} - \rho_{ga}\frac{\boldsymbol{g}_{sh,S}}{\|\boldsymbol{g}_{sh,S}\|\cdot\|\boldsymbol{g}_{sh,T}\|} + \rho_{gn}\frac{\boldsymbol{g}_{sh,T}}{\|\boldsymbol{g}_{sh,T}\|}, \boldsymbol{P}_T + \rho_{gn}\frac{\boldsymbol{g}_T}{\|\boldsymbol{g}_T\|}),$$

$$\mathcal{L}_S^{\text{PGA}}(\boldsymbol{P}) := \mathcal{L}_S(\boldsymbol{P}_{sh} - \rho_{ga}\frac{\boldsymbol{g}_{sh,T}}{\|\boldsymbol{g}_{sh,S}\|\cdot\|\boldsymbol{g}_{sh,T}\|} + \rho_{gn}\frac{\boldsymbol{g}_{sh,S}}{\|\boldsymbol{g}_{sh,S}\|}, \boldsymbol{P}_S + \rho_{gn}\frac{\boldsymbol{g}_S}{\|\boldsymbol{g}_S\|}).$$

As aforementioned, we use scalarization method, i.e. reweighting loss functions with $\lambda$ put on the PGA source objective. As a result, the PGA gradient updates for prompts are

$$\boldsymbol{g}_{sh,T}^{\text{PGA}}, \boldsymbol{g}_T^{\text{PGA}} := \nabla_{\boldsymbol{P}}\mathcal{L}_T^{\text{PGA}}(\boldsymbol{P}), \quad \boldsymbol{g}_{sh,S}^{\text{PGA}}, \boldsymbol{g}_S^{\text{PGA}} := \nabla_{\boldsymbol{P}}\mathcal{L}_S^{\text{PGA}}(\boldsymbol{P}),$$

$$\boldsymbol{P}_S = \boldsymbol{P}_S - \eta\boldsymbol{g}_S^{\text{PGA}}, \qquad\qquad \boldsymbol{P}_T = \boldsymbol{P}_T - \eta\boldsymbol{g}_T^{\text{PGA}},$$

$$\boldsymbol{P}_{sh} = \boldsymbol{P}_{sh} - \eta(\boldsymbol{g}_{sh,T}^{\text{PGA}} + \lambda\boldsymbol{g}_{sh,S}^{\text{PGA}}).$$

However, computing these PGA gradients will trigger the computation of the Hessian matrix. Hence, we approximate them with a practical version:

$$\boldsymbol{g}_{sh,T}^{\text{PGA}}, \boldsymbol{g}_T^{\text{PGA}} := \nabla_{\boldsymbol{P}}\mathcal{L}_T^{\text{PGA}}(\boldsymbol{P})$$

$$\approx \nabla_{\boldsymbol{P}}\mathcal{L}_T(\boldsymbol{P}_{sh}, \boldsymbol{P}_T)\big|_{\boldsymbol{P}_{sh}=\boldsymbol{P}_{sh}-\rho_{ga}\frac{\boldsymbol{g}_{sh,S}}{\|\boldsymbol{g}_{sh,S}\|\cdot\|\boldsymbol{g}_{sh,T}\|}+\rho_{gn}\frac{\boldsymbol{g}_{sh,T}}{\|\boldsymbol{g}_{sh,T}\|}, \boldsymbol{P}_T=\boldsymbol{P}_T+\rho_{gn}\frac{\boldsymbol{g}_T}{\|\boldsymbol{g}_T\|}},$$

$$\boldsymbol{g}_{sh,S}^{\text{PGA}}, \boldsymbol{g}_S^{\text{PGA}} := \nabla_{\boldsymbol{P}}\mathcal{L}_S^{\text{PGA}}(\boldsymbol{P})$$

$$\approx \nabla_{\boldsymbol{P}}\mathcal{L}_S(\boldsymbol{P}_{sh}, \boldsymbol{P}_S)\big|_{\boldsymbol{P}_{sh}=\boldsymbol{P}_{sh}-\rho_{ga}\frac{\boldsymbol{g}_{sh,T}}{\|\boldsymbol{g}_{sh,S}\|\cdot\|\boldsymbol{g}_{sh,T}\|}+\rho_{gn}\frac{\boldsymbol{g}_{sh,S}}{\|\boldsymbol{g}_{sh,S}\|}, \boldsymbol{P}_S=\boldsymbol{P}_S+\rho_{gn}\frac{\boldsymbol{g}_S}{\|\boldsymbol{g}_S\|}}.$$

## B.2 Extension to Multi-source UDA

Our method can be easily extended to work with multi-source domains by noting that the target gradient is aligned with each of the source gradients.

$$\boldsymbol{g}_{sh,T}^{\text{PGA}}, \boldsymbol{g}_T^{\text{PGA}} := \nabla_{\boldsymbol{P}}\mathcal{L}_T^{\text{PGA}}(\boldsymbol{P}),$$

$$\boldsymbol{g}_{sh,i}^{\text{PGA}}, \boldsymbol{g}_{S,i}^{\text{PGA}} := \nabla_{\boldsymbol{P}}\mathcal{L}_{S,i}^{\text{PGA}}(\boldsymbol{P}), \forall i = 1 \to N$$

$$\boldsymbol{P}_{S,i} = \boldsymbol{P}_{S,i} - \eta\boldsymbol{g}_{S,i}^{\text{PGA}}, \forall i = 1 \to N$$

$$\boldsymbol{P}_T = \boldsymbol{P}_T - \eta\boldsymbol{g}_T^{\text{PGA}},$$

$$\boldsymbol{P}_{sh} = \boldsymbol{P}_{sh} - \eta(\boldsymbol{g}_{sh,T}^{\text{PGA}} + \lambda\sum_i\boldsymbol{g}_{sh,i}^{\text{PGA}}),$$

$$\boldsymbol{g}_{sh,T}^{\text{PGA}}, \boldsymbol{g}_T^{\text{PGA}} \approx \nabla_{\boldsymbol{P}}\mathcal{L}_T(\boldsymbol{P}_{sh}, \boldsymbol{P}_T)\big|_{\boldsymbol{P}_{sh}=\boldsymbol{P}_{sh}-\rho_{ga}\sum_i\frac{\boldsymbol{g}_{sh,i}}{\|\boldsymbol{g}_{sh,i}\|\cdot\|\boldsymbol{g}_{sh,T}\|}+\rho_{gn}\frac{\boldsymbol{g}_{sh,T}}{\|\boldsymbol{g}_{sh,T}\|}, \boldsymbol{P}_T=\boldsymbol{P}_T+\rho_{gn}\frac{\boldsymbol{g}_T}{\|\boldsymbol{g}_T\|}}, \tag{28}$$

$$\boldsymbol{g}_{sh,i}^{\text{PGA}}, \boldsymbol{g}_{S,i}^{\text{PGA}} \approx \nabla_{\boldsymbol{P}}\mathcal{L}_{S,i}(\boldsymbol{P}_{sh}, \boldsymbol{P}_{S,i})\big|_{\boldsymbol{P}_{sh}=\boldsymbol{P}_{sh}-\rho_{ga}\frac{\boldsymbol{g}_{sh,T}}{\|\boldsymbol{g}_{sh,i}\|\cdot\|\boldsymbol{g}_{sh,T}\|}+\rho_{gn}\frac{\boldsymbol{g}_{sh,i}}{\|\boldsymbol{g}_{sh,i}\|}, \boldsymbol{P}_{S,i}=\boldsymbol{P}_{S,i}+\rho_{gn}\frac{\boldsymbol{g}_{S,i}}{\|\boldsymbol{g}_{S,i}\|}}. \tag{29}$$

The details of our proposed method for the general case of $N$ source domains are presented in Algorithm 1. When $N = 1$, our method degrades to PGA.

# C Experimental details

In this section, we provide additional information for our experimental settings in Section C.1 and C.2 then include detailed ablation studies and other empirical results in Section C.3.

---

**Algorithm 1** Prompt gradient alignment for unsupervised domain adaptation

---

**Input:** Prompt $\boldsymbol{P} = [\boldsymbol{P}_{sh}, \{\boldsymbol{P}_{S,i}\}_{i=1}^N, \boldsymbol{P}_T]$, gradient norm penalization trade-off $\rho_{\text{gn}}$, alignment strength $\rho_{\text{ga}}$, source-gradient trade-off $\lambda$, learning rate $\eta$.
**Output:** Updated prompt $\boldsymbol{P}^*$

1: Compute target loss $\mathcal{L}_T(\boldsymbol{P}_{sh}, \boldsymbol{P}_T)$ as in Eq. 3
2: Compute gradients of shared and target-specific prompts w.r.t target loss
$$\boldsymbol{g}_{sh,T}, \boldsymbol{g}_T \leftarrow \nabla_{\boldsymbol{P}} \mathcal{L}_T(\boldsymbol{P}_{sh}, \boldsymbol{P}_T)$$
3: Compute source losses $\mathcal{L}_{S,i}(\boldsymbol{P}_{sh}, \boldsymbol{P}_{S,i})$ as in Eq. 2
4: Compute gradient of shared and source-specific prompts w.r.t each source loss
$$\boldsymbol{g}_{sh,i}, \boldsymbol{g}_{S,i} \leftarrow \nabla_{\boldsymbol{P}} \mathcal{L}_{S,i}(\boldsymbol{P}_{sh}, \boldsymbol{P}_{S,i}), \forall i = 1 \rightarrow N$$
5: Compute $\boldsymbol{g}_{sh,T}^{\text{PGA}}, \boldsymbol{g}_T^{\text{PGA}}$ as in Eq. 28
6: Compute $\boldsymbol{g}_{sh,i}^{\text{PGA}}, \boldsymbol{g}_{S,i}^{\text{PGA}}$ as in Eq. 29 $\forall i = 1 \rightarrow N$
7: Compute combined gradient of shared prompt $\boldsymbol{g}_{sh}^{\text{PGA}} = \boldsymbol{g}_{sh,T}^{\text{PGA}} + \lambda \sum_i \boldsymbol{g}_{sh,i}^{\text{PGA}}$
8: Update prompt
$$\boldsymbol{P}^* = [\boldsymbol{P}_{sh}, \{\boldsymbol{P}_{S,i}\}_{i=1}^N, \boldsymbol{P}_T] - \eta[\boldsymbol{g}_{sh}^{\text{PGA}}, \{\boldsymbol{g}_{S,i}^{\text{PGA}}\}_{i=1}^N, \boldsymbol{g}_T^{\text{PGA}}]$$

---

## C.1 Datasets

ImageCLEF is a small-scaled dataset with 1,800 images across 12 object categories from three domains: ImageNet ILSVRC 2012 (I), Pascal VOC 2012 (P), and Caltech-256 (C). Office-Home is a medium-scaled dataset containing approximately 15,500 images from 65 categories in four domains: Art, Clipart, Product, and Real World. DomainNet is the largest dataset, comprising around 600,000 images from 345 categories across six domains: Clipart, Infograph, Painting, Quickdraw, Real, and Sketch.

## C.2 Implementation details

For fair comparisons, we use a ResNet50 as our backbone on Image-CLEF and Office-Home and a ResNet101 on DomainNet. Their weights are taken from pretrained-CLIP and kept frozen during training. Prompts are trained with the mini-batch SGD optimizer with a learning rate of 0.003 and 0.005. We use a batch size of 32 and adopt a cosine learning rate scheduler. For hyper-parameters, token lengths $M_1$ and $M_2$ are both set to 16. Pseudo-label threshold $\tau$ is set to 0.4 for producing reliable labels. $\rho_{gn}$, $\rho_{ga}$ and $\lambda$ are found using grid-search. Details are provided in the public source code.

During inference, we average the prediction of both source $\boldsymbol{P}_S$ and target $\boldsymbol{P}_T$ prompts, which empirically yield the best performance. Please note that the inference cost remains almost the same as using a pretrained CLIP as computing class embeddings is an one-time-cost. The complexity grows linearly with the number of prompts during training ($= 2$ with PGA and $N + 1$ in the case of MPGA), which is typically not a big issue in practice since the model training can quickly converge by fine-tuning under intrinsic dimension [119]. We further confirm this in the computation complexity ablation study below.

## C.3 Additional experiments

### C.3.1 Illustrative example

We run a small multi-objective-optimization problem on the ZDT-1 problem [120]. The ZDT-1 problems have a 30-dimensional variable and two differentiable objective functions $f_1, f_2$:

$$\min f_1(x)$$
$$\min f_2(x) = g(x)h\left(f_1(x), g(x)\right)$$

The function $g(x)$ can be considered as the function for convergence, their formulas are given by:

$$f_1(x) = x_1$$

$$g(x) = 1 + \frac{9}{n-1} \sum_{i=2}^{n} x_i$$

$$h(f_1, g) = 1 - \sqrt{f_1/g}$$

$$0 \le x_i \le 1 \quad i = 1, \dots, n$$

with the Pareto solutions are given by:

$$0 \le x_1^* \le 1 \quad \text{and} \quad x_i^* = 0 \text{ for } i = 2, \dots, n$$

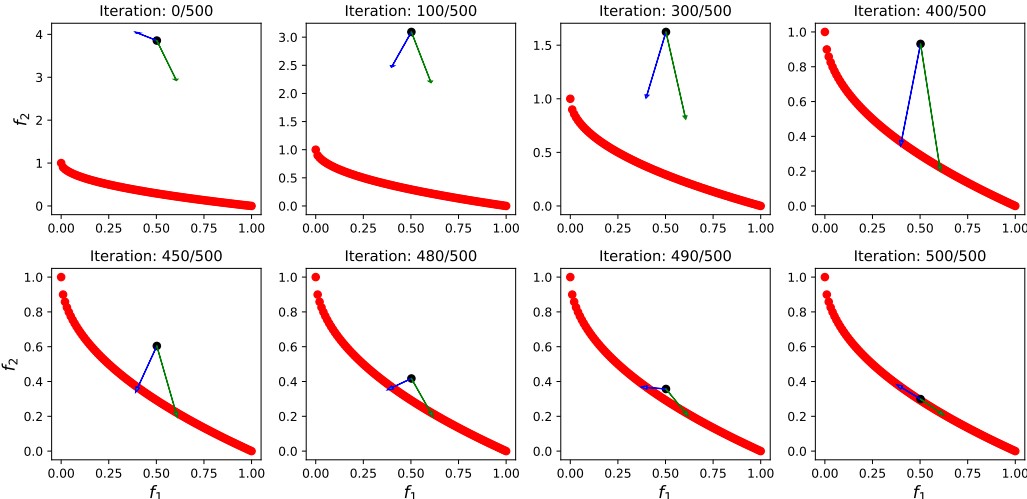

Figure 4: ZDT-1 task-specific gradient directions at different iterations. Red curve represents the Pareto front while the blue and green arrows indicate the updating directions for minimizing $f_1$ and $f_2$, respectively.

As can be seen from Figure 4, the cosine similarity increases at the beginning of the training and then decreases when the obtained solution reach the region near the Pareto front. This behavior aligns with the gradient similarity evolution experiment in the main paper.

### C.3.2 Large-scale single-source unsupervised domain adaptation

Apart from those experiments in the main paper, we expand the single-source unsupervised domain adaptation setup by including the empirical results on two large-scale synthetic-to-real benchmark for classification adaptation S2RDA-49 and S2RDA-MS-39 [121]. For each task, synthetic samples are created by rendering 3D models from ShapeNet, matching the label space of the real/target domain, with 12K RGB images per class. The S2RDA-49 real domain contains 60,535 images across 49 classes from various sources including the ImageNet validation set. The S2RDA-MS-39 real domain includes 41,735 natural images for 39 classes from MetaShift, featuring complex contexts like object co-occurrence and attributes, which adds to the task's difficulty.

Table 6: Unsupervised domain adaptation results on S2RDA. The best accuracy is indicated in **bold**.

| Transfer Task | No Adaptation | | DANN | | MCD | | RCA | | SRDC | | DisClusterDA | | CLIP | | DAPL | | PGA (Ours) | |
|---|---|---|---|---|---|---|---|---|---|---|---|---|---|---|---|---|---|---|
| | Acc. | Mean | Acc. | Mean | Acc. | Mean | Acc. | Mean | Acc. | Mean | Acc. | Mean | Acc. | Mean | Acc. | Mean | Acc. | Mean |
| S2RDA-49 | 51.9 | 42.2 | 47.1 | 47.6 | 42.5 | 47.8 | 47.1 | 48.5 | 61.5 | 53.0 | 53.0 | 52.3 | 69.9 | 65.7 | 71.5 | 66.5 | **74.1** | **67.8** |
| S2RDA-MS-39 | 22.0 | 20.5 | 22.8 | 22.2 | 22.1 | 22.2 | 23.3 | 22.5 | 25.8 | 24.6 | 27.1 | 25.3 | 36.4 | 35.8 | 36.9 | 35.7 | **38.0** | **36.9** |

Table 6 illustrates accuracy and mean score over classes, where utilizing pretrained vision-language models still shows their impressive performance. Using pretrained CLIP standalone outperforms other traditional DA methods and PGA further boosts the performance by large margins, $4\%$ on S2RDA-49 and $1.5\%$ on S2RDA-MS-39, respectively.

### C.3.3 Ablation studies

Similar to previous work on CLIP adaptation[25, 28], we vary the pseudo label threshold $\tau$ value to study its sensitivity. As can be seen in Figure 7, both PGA and MPGA's performance is relatively stable across different values of $\tau$, indicating that our methods are not sensitive to $\tau$, and the best result is obtained at a reasonable trade-off between the quantity and quality of pseudo data.

Table 7: Accuracy (%) of different threshold $\tau$ on ImageCLEF.

|      | 0.1  | 0.2  | 0.3  | 0.4  | 0.5  | 0.6  | 0.8  | 0.9  |
|------|------|------|------|------|------|------|------|------|
| PGA  | 91.5 | 92.0 | 92.1 | 92.4 | 92.3 | 91.1 | 92.0 | 92.0 |
| MPGA | 92.4 | 92.6 | 92.9 | 92.7 | 92.7 | 92.7 | 92.6 | 92.5 |

In Figure 5, we provide the complexity for some comparative baselines. Accuracy curve (left): While DANN and CDAN obtain their best performance at approximately 77% after more than 1000s, PGA and MPGA achieve 84% within 100s. Besides, the first stage of pairwise source-target training of MPA takes 159s, followed by 35s for the second stage to actually train the final model. Number of Trainable Parameters (middle): PGA and MPGA, with fewer than 140k parameters, require significantly fewer parameters than MPA, DANN and CDAN, which have around 1M, 48.9M and 51.7M parameters, respectively. GPU Memory Usage (right) PGA, MPGA, and MPA exhibit substantially lower memory footprints, around 1300MB compared to 7000MB of DANN and CDAN throughout training.

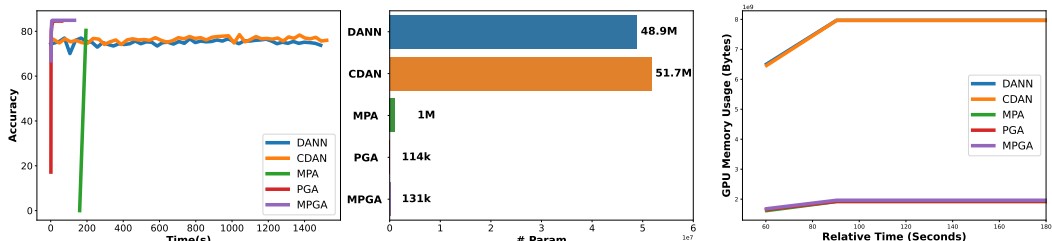

Figure 5: Computational complexity: accuracy curve (left), number of trainable parameters (middle), and GPU memory (right).

Figure 6 shows that PGA is generally not sensitive to $\rho_{ga}$ and $\rho_{gn}$ within their acceptable range, i.e. 1e-2 to 10 for $\rho_{ga}$ and 1e-5 to 0.1 for $\rho_{gn}$. Specifically, (i) a too large value of $\rho_{gn}$ is less effective than smaller ones; (ii) ImageCLEF prefers larger values of $\rho_{ga}$ while OfficeHome prefers smaller ones, suggesting that source and target domains in the former dataset may be more similar than those in the latter, hence over-matching gradients in the latter dataset may be adverse.

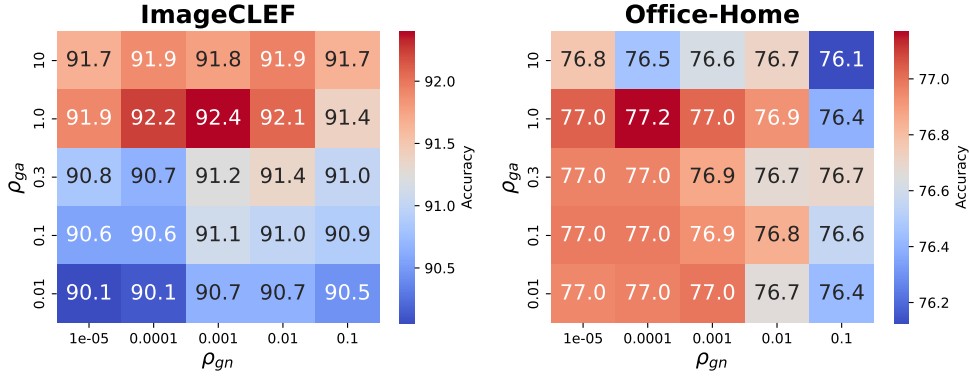

Figure 6: Parameter sensitivity analysis on $\rho_{gn}$ and $\rho_{ga}$ of PGA on ImageCLEF and Office-Home with CLIP-RN50 backbone.

We present results of our methods using ViT-B/16, ViT-L/14 backbones on OfficeHome in Tables 8 and 9, following experimental setups in [122, 123]. We can observe the superiority of our methods among all baselines while finetuning a small portion of the backbones using prompt tuning. Especially, PGA outperforms the second-best method on ViT-B/16 backbone by $\approx 1\%$ accuracy score.

Table 8: Accuracy (%) on Office-Home of ViT-based vision encoder CLIP backbones (except CDTrans* uses DeiT). The overall best accuracy and best within per backbone are indicated in **bold** and underline, respectively.

| Method | Backbone | Ar→Cl | Ar→Pr | Ar→Rw | Cl→Ar | Cl→Pr | Cl→Rw | Pr→Ar | Pr→Cl | Pr→Rw | Rw→Ar | Rw→Cl | Rw→Pr | Avg |
|---|---|---|---|---|---|---|---|---|---|---|---|---|---|---|
| CDTrans* | | 68.8 | 85.0 | 86.9 | 81.5 | 87.1 | 87.3 | 79.6 | 63.3 | 88.2 | 82.0 | 66.0 | 90.6 | 80.5 |
| TVT | | 74.9 | 86.8 | 89.5 | 82.8 | 88.0 | 88.3 | 79.8 | 71.9 | 90.1 | 85.5 | 74.6 | 90.6 | 83.6 |
| linear probe CLIP | | 60.1 | 73.7 | 80.9 | 66.4 | 76.4 | 76.3 | 63.4 | 61.0 | 82.3 | 74.7 | 64.8 | 83.3 | 72.4 |
| CoOp | | 70.0 | 90.8 | 90.9 | 83.2 | 90.9 | 89.2 | 82.0 | 71.8 | 90.5 | 83.8 | 71.5 | 92.0 | 83.9 |
| CoCoOp | | 70.4 | 91.4 | 90.4 | 83.5 | 91.8 | 90.3 | 83.4 | 70.9 | 91.0 | 83.4 | 71.6 | 91.7 | 84.1 |
| VPT-shallow | | 66.9 | 89.1 | 89.1 | 81.7 | 89.0 | 89.2 | 81.6 | 70.0 | 89.1 | 81.7 | 66.9 | 89.0 | 81.7 |
| VPT-deep | ViT-B/16 | 71.6 | 89.9 | 90.3 | 82.8 | 91.0 | 89.7 | 82.0 | 71.5 | 90.3 | 84.6 | 71.7 | 91.6 | 83.9 |
| IVLP | | 71.4 | 91.7 | 90.8 | 83.6 | 90.2 | 89.3 | 82.2 | 72.4 | 90.4 | 84.1 | 72.1 | 92.0 | 84.2 |
| MaPLe | | 72.2 | 91.6 | 90.3 | 82.6 | 90.9 | 89.8 | 82.4 | 71.6 | 90.0 | 85.1 | 72.0 | 92.1 | 84.2 |
| CLIP | | 67.8 | 89.0 | 89.8 | 82.9 | 89.0 | 89.8 | 82.9 | 67.8 | 89.8 | 82.9 | 67.8 | 89.0 | 82.4 |
| DAPL | | 70.6 | 90.2 | 91.0 | 84.8 | 89.2 | 90.9 | 84.8 | 70.5 | 90.6 | 84.8 | 70.1 | 90.8 | 84.0 |
| PGA (Ours) | | 71.8 | 91.5 | 91.0 | 84.8 | 91.6 | 90.9 | 84.9 | 71.5 | 91.1 | 85.9 | 72.1 | 92.4 | 85.1 |
| CLIP | | 74.2 | 93.1 | 93.3 | 87.3 | 93.1 | 93.3 | 87.3 | 74.2 | 93.3 | 87.3 | 74.2 | 93.1 | 87.0 |
| DAPL | ViT-L/14 | 77.3 | 94.6 | **94.3** | 88.6 | 94.6 | 94.0 | 88.8 | 76.8 | 94.0 | **89.0** | 77.8 | 94.4 | 88.7 |
| PGA (Ours) | | **79.0** | **95.1** | **94.3** | **88.9** | **95.1** | **94.2** | **88.9** | **78.8** | **94.2** | 88.9 | **79.0** | **95.3** | **89.4** |

Following a different protocol, Table 9 provides the results of ViT-L/14 backbones on Office-Home but with three source domains per category on Art, Clipart, Realworld and Product domain. In this setup, MPGA and PGA still consistently yield the best and second-best scores among all categories.

Table 9: Three-source domain adaptation of the Office-Home dataset on ViT-L/14.

| Method | → Ar | → Rw | → Pr | Avg |
|---|---|---|---|---|
| CLIP ZS(G) | 84.97 | 91.94 | 90.96 | 89.29 |
| CLIP ZS(A) | 86.34 | 92.10 | 87.73 | 88.73 |
| CLIP LP | 87.02 | 92.55 | 92.70 | 90.76 |
| LADS | 87.71 | 93.86 | 93.00 | 91.52 |
| LanDA | 88.83 | 94.09 | 93.22 | 92.05 |
| PGA (Ours) | 89.17 | 95.37 | 94.34 | 92.96 |
| MPGA (Ours) | **89.88** | **95.49** | **94.97** | **93.45** |

### C.3.4   Domain adaptation with label shift

This section is to study how does the method performs when there are extreme label distribution shifts between source and target domains. We test PGA on the setting of label shift following [124], where the source or target domains are down-sampled with only $30\%$ of data from the first-half of the classes are taken (indicated by s- prefix).

Table 10: Accuracy (%) on the sub-sampled Office-Home for unsupervised domain adaptation. The prefix s- denotes the domain where we sample only 30% of the images from the first half of its classes, following the label shift setting from prior work.

| Method | sAr→Cl | sAr→Pr | sAr→Rw | sCl→Ar | sCl→Pr | sCl→Rw | sPr→Ar | sPr→Cl | sPr→Rw | sRw→Ar | sRw→Cl | sRw→Pr | Avg |
|---|---|---|---|---|---|---|---|---|---|---|---|---|---|
| ResNet-50 | 35.7 | 54.7 | 62.6 | 43.7 | 52.5 | 56.6 | 44.3 | 33.0 | 65.2 | 57.1 | 40.5 | 70.0 | 51.4 |
| DANN | 36.1 | 54.2 | 61.7 | 44.3 | 52.6 | 56.4 | 44.6 | 37.1 | 65.2 | 56.7 | 43.2 | 69.9 | 51.8 |
| JAN | 34.5 | 56.9 | 64.5 | 46.2 | 56.8 | 59.0 | 50.6 | 37.2 | 70.0 | 58.7 | 40.6 | 72.0 | 53.9 |
| CDAN | 38.9 | 56.8 | 64.8 | 48.0 | 60.0 | 61.2 | 49.7 | 41.4 | 70.2 | 62.4 | 47.0 | 74.7 | 56.3 |
| IWDANN | 39.8 | 63.0 | 68.7 | 47.4 | 61.1 | 60.4 | 50.4 | 41.6 | 72.5 | 61.0 | 49.4 | 76.1 | 57.6 |
| IWJAN | 36.2 | 61.0 | 66.3 | 48.7 | 59.9 | 61.9 | 52.9 | 37.7 | 70.9 | 60.3 | 41.5 | 73.3 | 55.9 |
| IWCDAN | 43.0 | 65.0 | 71.3 | 52.9 | 64.7 | 66.5 | 54.9 | 44.8 | 75.9 | 67.0 | 50.5 | 78.6 | 61.2 |
| PCT | 51.9 | 69.7 | 76.5 | 63.3 | 70.8 | 71.1 | 66.0 | 49.9 | 82.0 | 73.1 | **58.6** | 83.2 | 67.8 |
| PGA (Ours) | **54.7** | **85.4** | **85.4** | **75.3** | **84.4** | **85.2** | **75.4** | **54.9** | **85.7** | **75.6** | 54.3 | **85.7** | **75.2** |

Label-shift results presented in Table 10 and 11 below and Table 5 in the main text show the effectiveness of PGA on different levels of label shift. PGA consistently yields superior performance on every sub-experiment under these two setups.

Table 11: Accuracy (%) on the sub-sampled (target) Office-Home for unsupervised domain adaptation.

| Method | Ar→sCl | Ar→sPr | Ar→sRw | Cl→sAr | Cl→sPr | Cl→sRw | Pr→sAr | Pr→sCl | Pr→sRw | Rw→sAr | Rw→sCl | Rw→sPr | Avg |
|---|---|---|---|---|---|---|---|---|---|---|---|---|---|
| ResNet-50 | 41.5 | 65.8 | 73.6 | 52.2 | 59.5 | 63.6 | 51.5 | 36.4 | 71.3 | 65.2 | 42.8 | 75.4 | 58.2 |
| DANN | 47.8 | 55.9 | 66.0 | 45.3 | 54.8 | 56.8 | 49.4 | 48.0 | 70.2 | 65.4 | 55.5 | 72.7 | 58.3 |
| JAN | 45.8 | 69.7 | 74.9 | 53.9 | 63.2 | 65.0 | 56 | 42.5 | 74 | 65.9 | 47.4 | 78.8 | 61.4 |
| CDAN | 51.1 | 69.7 | 74.6 | 56.9 | 60.4 | 64.6 | 57.2 | 45.5 | 75.6 | 68.5 | 52.7 | 79.8 | 63.0 |
| IWDANN | 48.7 | 62.0 | 71.6 | 50.4 | 57.0 | 60.3 | 51.4 | 41.1 | 69.9 | 62.6 | 51.0 | 77.2 | 58.6 |
| IWJAN | 44.0 | 71.9 | 75.1 | 55.2 | 65.0 | 67.7 | 57.1 | 42.4 | 74.9 | 66.1 | 46.1 | 78.5 | 62.0 |
| IWCDAN | 52.3 | 72.2 | 76.3 | 56.9 | 67.3 | 67.7 | 57.2 | 44.8 | 77.8 | 67.3 | 53.3 | 80.6 | 64.6 |
| PCT-Uniform | 55.8 | 77.6 | 80.4 | 65.1 | 72.3 | 74.7 | 67.0 | 50.9 | 81.1 | 72.6 | 57.0 | 84.0 | 69.8 |
| PCT-Learnable | **57.5** | 78.2 | 80.5 | 66.7 | 74.3 | 75.4 | 64.6 | 50.7 | 81.3 | 72.9 | 57.3 | 83.5 | 70.2 |
| PGA (Ours) | 57.4 | **84.8** | **86.4** | **76.0** | **84.6** | **85.6** | **74.5** | **57.1** | **86.1** | **75.9** | **57.4** | **85.3** | **75.9** |

## C.4    Training Resources

All experiments are run on Intel(R) Xeon(R) Platinum 8358 CPU @ 2.60GHz and NVIDIA A100-SXM4-80GB GPU.

# D    Additional related work

Another work sharing the same intuition of gradient alignment is ProGrad [125], which manipulates gradient of the fine-tuned loss to preserve general knowledge of the pretrained model. Similar to other gradient-based MTL methods [34, 35], it attempts to remove conflicts between per-objective gradients at each time step, thus is orthogonal to our approach. In contrast, we aim to stimulate their inherent consensus throughout training by encouraging the same training trajectory for both domains, hence, the model can find commonly good regions for them. Another concept that relates to gradient alignment is meta-learning. This has been introduced to Domain generalization in [126, 127]. Their intuition is a training procedure that enables the model to achieve low loss on a subset of training domains after having learned the other ones, and they work on the full model space. In a recent work about Vision-Language Models [128], meta-learning was used to deal with the problem of few-shot prompt learning by meta-learning prompt initialization. The gradient of the inner loop is modified with a learnable regulating function, and data for the support and query sets are found by hierarchically clustering an auxiliary large-scale image-text dataset. This method also has the impact of aligning gradient between support and query data as a result of meta-learning. However, its computation and space complexity is rather large as it requires the computation of Hessian matrix, web-scale of image-text pairs, and meta-learns the soft initialization for prompts.

# E    Limitations and Future works

First, our work relies on pretrained-CLIP, meaning that if UDA data is too different from pretrained knowledge, our method may fail to learn adequately. Therefore adapting our method to scratch-training scenarios without heavy computation and space complexity should be investigated. Second, the derived bound can be potentially loose as the number of training iterations increases. Thus studying other types of bounds could be an interesting work. Finally, as we mentioned, a strategy to explicitly align feature distribution across domains is worth looking into.

