# OpenReview forum: "Enhancing Domain Adaptation through Prompt Gradient Alignment"
_NeurIPS.cc/2024/Conference — NeurIPS 2024 poster_

### Official Review · Reviewer_aPJr · 2024-07-03

**Soundness:** 3
**Presentation:** 3
**Contribution:** 3
**Rating:** 6
**Confidence:** 3

**Summary:**

This paper aims to leverage prompt tuning of vision-language models for Unsupervised Domain Adaptation (UDA) tasks. The authors formulate UDA as a multi-objective problem where each objective is modeled by a domain loss. To resolve conflicts between domains, they manipulate gradients by maximizing their cosine similarity. Additionally, to stabilize the training procedure, they propose using the norm of the gradients as a regularization term. They also provide a generalization error bound for their method. Empirical results demonstrate the effectiveness of the proposed method.

**Strengths:**

The paper is well-organized and the presentation is clear. To my knowledge, the idea of aligning gradients of different objectives by maximizing cosine similarity and regularizing the gradient norm is intuitive and novel. The authors also provide a generalization bound for the proposed method, which is a valuable contribution.

**Weaknesses:**

Major:

(1) Recent empirical and theoretical studies [1] show that simply reweighting the loss of different objectives can match the performance of gradient surgery. When the model is under-parameterized (which may be the case for prompt learning, as the prompt parameters are relatively small compared to the CLIP parameters), simple reweighting is sufficient to fully explore the Pareto front. My question is why the proposed gradient manipulation is better. Can the authors provide theoretical analysis or empirical validation?

[1] Revisiting Scalarization in Multi-Task Learning: A Theoretical Perspective

(2) The backbone CLIP-ResNet-50 is a weak backbone. Please consider conducting some experiments on a CLIP-ViT based model.

Minor:

(1) Missing references. There have been some studies that use gradient alignment for prompt learning, e.g., [3,4]. Please consider adding them to the related works section.

[3] Prompt-aligned Gradient for Prompt Tuning
[4] Gradient-Regulated Meta-Prompt Learning for Generalizable Vision-Language Models

(2) Table 1 in the Introduction is not well explained. It does not specify which dataset and CLIP backbone are used, and the symbols $\rightarrow C, I, P$ are not explained. In addition, Table 2 should be referenced as Figure 1.

**Questions:**

Please refer to the weaknesses, and I am willing to raise the score if the major weaknesses are addressed.

**Limitations:**

Limitations have been discussed and no negative societal impact has been identified.

---

> ### Author Rebuttal · Authors · 2024-08-06
>
> >1. why the proposed gradient manipulation is better?:
>
> Thank you for your insightful question. We would appreciate further clarification on your reference to the paper discussing the limitations of scalarization, particularly in under-parametrized setups, as it seems to suggest scalarization’s inherent inability to fully explore the solution space.
> - Regarding your point of the efficiency of scalarization over multi-task learning (MTL) methods, we discuss this in lines 53-59 of the main text. In particular, we simply re-weight per-task gradients (similar to scalarization) instead of adopting multi task learning methods. In addition, the MTL methods in [33,92] are generic methods and are designed to optimize all objectives simultaneously. In contrast, we focus more on adapting our model to the target task rather than training a model that has universal performance across all domains or profiling the entire Pareto front. This further motivates our choice to use re-weighting source and target objectives' gradients.
>  - While studies [33, 92] demonstrate scalarization’s advantages over certain MTL methods like IMTL, MGDA, GradDrop, and PCGrad (ProGrad), they do not establish scalarization’s equivalence with all gradient manipulation techniques. In contrast, gradient alignment has been proven effective in learning invariant features [69], and penalizing gradient norms is widely recognized for enhancing model generalization [102, 18, 4, 90]. Empirically, we conduct some ablation studies on these two components in Table 6, the illustrative example, and gradient similarity experiments, to show that solely using scalarization is not enough.
>
> 2. We provide results of our methods using ViT-B/16, ViT-L/14 backbones on OfficeHome in Tables 3 and 5 in the attached PDF, following experimental setups in [R1, R2]. We can observe the superiority of our methods among all baselines while finetuning a small portion of the backbones.
>
> 3. Thank you for the suggested references. We will add them in the revision.
>
> 4. Caption for Table 1 and reference typos: Thank you for pointing these out, we will fix them in the revision. The experiment in Table 1 is on dataset ImageCLEF with I, P, C representing ImageNet ILSVRC 2012, Pascal VOC 2012, and Caltech-256, respectively, and ResNet50 backbone is adopted. The results in Table 1 in fact are from Table 3, we will add detailed discription for this experiment in the later revision.
>
> We hope our response could help address your concerns, we are very open to further discussion.
>
> [R1] Wang, Zhenbin, et al. "Landa: Language-guided multi-source domain adaptation.", arXiv 24
>
> [R2] Singha, Mainak, et al. "Ad-clip: Adapting domains in prompt space using clip.", ICCV 23

---

> > ### Comment · Reviewer_aPJr · 2024-08-10
> >
> > Thanks for the response. All my concerns are resolved and I have raised my rating

---

> > > ### Author Response · Authors · 2024-08-11
> > > **Official Comment by Authors**
> > >
> > > We sincerely thank you for your insightful review and supportive feedback to make the paper more complete. Your support is highly appreciated, and we are glad that our responses have addressed your concerns.

---

### Official Review · Reviewer_dhjM · 2024-07-09

**Soundness:** 3
**Presentation:** 2
**Contribution:** 3
**Rating:** 7
**Confidence:** 3

**Summary:**

This paper proposed a novel domain adaptation method that tunes the text prompts based on self-training with pseudo labels. The proposed method treats the source training and target training as multi-objective optimization problems, and it introduces to alignment of the gradients from both training (Prompt Gradient Alignment) so as not to cause conflict between the source and target training. The authors validated the proposed method empirically and theoretically.

**Strengths:**

+ This paper is rich in technical novelty.
  + There are many methods in domain generalization research that aim to align gradients from multiple domains. However, requires heavy hessian calculations. The proposed method successfully avoids this issue by approximation with Taylor expansion.
+ The proposed method is empirically and theoretically justified.

**Weaknesses:**

+ It is questionable whether the comparison with existing methods is fair.
  + The old domain adaptation methods fail to show their effectiveness when they are applied to the CLIP model. I suspect that those domain adaptation methods were not properly applied to CLIP. Is there any reason why they fail?
  + Conversely, would it not be possible to evaluate the proposed method in accordance with existing domain adaptation benchmarks? For example, would it be possible to evaluate the proposed method by updating the model parameters instead of the prompts?
+ There are some additional related works that should be discussed in the paper.
  + Bose, Shirsha, et al. "Stylip: Multi-scale style-conditioned prompt learning for clip-based domain generalization." Proceedings of the IEEE/CVF Winter Conference on Applications of Computer Vision. 2024. This work proposes a prompt training method for the domain generalization tasks.
  + Zhu, Beier, et al. "Prompt-aligned gradient for prompt tuning." Proceedings of the IEEE/CVF International Conference on Computer Vision. 2023. This introduces a gradient matching idea into prompt tuning, which has a similar mind to this paper.
+ Some of the formulas are a little difficult to follow.
  + It is very difficult to understand which of eq 9 or eq 11 is the actual loss function. This goes for eq 16 and eq 18 as well. I recommend separating the formulations for easier understanding.

**Questions:**

+ As mentioned in the third bullet of weaknesses, I could not fully understand the formulations. Could you explain which is the actual loss function?

**Limitations:**

+ I did not found any additional limitation for this work.

---

> ### Author Rebuttal · Authors · 2024-08-06
>
> > whether the comparison with existing methods is fair
>
> As we follow the experimental settings in [6, 15, 22, R4, R5, R6], we use their reported results which we believe have been verified to be fair when comparing old UDA methods and prompt-based methods. Specifically, they share the same vision backbone as prompt-based ones, e.g. ResNet, and are optimized to reach their full potential. Only one advantage of prompt-based methods is the use of additional textual information about class names when making predictions. However, this comparison setup is not only used in UDA, but also adopted in many works in other domains such as spurious debiasing [R7], continual learning [R1, R2]. Therefore, we believe that old UDA baselines are fairly compared. Additionally, we include comparisons with more recent CLIP-based baselines in Table 5 of the attached PDF. Here we finetune only a small portion of the model compared to other baselines, but still obtain competitive performance.
> >is there any reason why they fail?
>
> Their underperformance could be attributed to the difficulty of using a single backbone to simultaneously learn domain-invariant and class discriminative features [22, 75]. This difficulty usually arises from the inherent entanglement between domain and semantic information [R3], which can be further amplified as more source domains are involved [6]. In contrast, we not only devote a set of shared prompts to learn domain-invariant features, but also domain-specific prompts to capture useful information for classification. By leveraging these prompts, we create a more meaningful representation, leading to more precise predictions for the target domain
> > would it be possible to evaluate the proposed method by updating the model
>
> Yes, it is possible. Our proposed algorithm and theory development are generic and could be extended to finetuning the entire model. However, this would still entail a large computation and space complexity due to the size of the whole model, despite the avoidance of Hessian matrix calculation.
> Even so, adapting the full model on source/target domain can easily lead to overfitting and potentially inferior results in the presence of domain shift, even in the case of transformer models [1, 14, 34]. Therefore we opt for prompt learning which is a better way to adapt CLIP to downstream tasks (similar to other CLIP-based baselines) and is less likely to overfit.
>
> > Related works
>
> Thanks for the suggestion, we will add them in the revision. Briefly, the second work, ProGrad, manipulates gradient of the fine-tuned loss to preserve general knowledge of the pretrained model. Although it shares the intuition of gradient alignment with our work, there is a significant difference: ProGrad attempts to remove conflicts between per-objective gradients at each time step, similar other gradient-based MTL methods such as [96, 39]. In contrast, we aim to stimulate their inherent consensus throughout training by encouraging the same training trajectory for both domains, hence, the model can find commonly good regions for them. The first work appears unrelated to ours as it approaches UDA from a architecture-view, i.e. introducing additional modules to generate content- and style-aware prompts, whereas we tackle it from the view of model optimization.
>
> > Difficulty of some formulas and the actual loss function
>
>  - Ideally, Eq.(9) and Eq.(16) would be the actual loss functions that we want to take derivative. The reason we approximate them to get Eq.(11) and Eq.(18) is to show that minimizing them can fulfill our purposes of (i) aligning prompt gradients and (ii) penalizing prompt gradient norms.
>  - Note that Eq.(16) indicates our full algorithm while Eq.(9) is only about gradient alignment.
>  - However, one difficulty in taking derivative of Eq.(9) or Eq.(16) is the computation of Hessian matrix, hence we devise a practical approximation as shown in Eq.(13) and Eq.(19), which allows us to only need to compute gradient of the original source/target loss, i.e. Eqs.(3,4), at the perturbed prompts.
>   - Please refer to the general response to see the actual loss function.
>
> [R1] Wang, Yabin, et al. "S-prompts learning with pre-trained transformers: An occam’s razor for domain incremental learning.", NeurIPS 22
>
> [R2] Yu, Jiazuo, et al. "Boosting continual learning of vision-language models via mixture-of-experts adapters.", CVPR 24
>
> [R3] Cai, Ruichu, et al. "Learning disentangled semantic representation for domain adaptation." IJCAI 19
>
> [R4] Bai, Shuanghao, et al. "Prompt-based distribution alignment for unsupervised domain adaptation." AAAI 24
>
> [R5] Li, Xinyao, et al. "Split to Merge: Unifying Separated Modalities for Unsupervised Domain Adaptation." CVPR 24
>
> [R6] Du, Zhekai, et al. "Domain-agnostic mutual prompting for unsupervised domain adaptation." CVPR 24
>
> [R7] Phan, Hoang, et al. "Controllable Prompt Tuning For Balancing Group Distributional Robustness", ICML 24

---

> > ### Comment · Reviewer_dhjM · 2024-08-10
> >
> > Thank you for the responses. I think my concerns have mostly been addressed. I still have concerns about the performance of the old domain adaptation methods, but I understand that is not a unique weakness of this paper but applied to all prior studies. I acknowledge that the proposed method is superior to the newer domain adaptation methods (e.g., DAPL) on equal conditions, so this concern may not be so critical. In conclusion, I maintain my positive evaluation for this paper.

---

> ### Author Response · Authors · 2024-08-11
> **Author response to Reviewer dhjM**
>
> We would like to thank the reviewer for acknowledging our effort and we are encouraged that our responses have addressed most of your concerns. Regarding your last concern about the fairness of our experiments with traditional UDA methods, we have examined DANN and CDAN using a CLIP backbone to provide more insight into the limitations of domain-invariant feature learning in adapting pretrained CLIP to new domains. We will run more experiments on other datasets and incorporate obtained results to the revision. It is important to note that our baseline results, so far, are taken directly from prior work, as we adhere to the recent protocols for adapting CLIP from prior work.
>
> | Method        | Backbone     | →C     | →I     | →P     | Average | # Param |
> |-|-|-|-|-|-|-|
> | SOURCE ONLY   | RN50-FC      |  94.7 | 90.2 | 79.3   | 88.1   | 38M     |
> | DANN          | RN50-FC      | 96.0 | 92.3 | 80.2   | 89.5   | 38M     |
> | CDAN          | RN50-FC      | 96.2 | 92.0 | 80.4   | 89.5  | 38M     |
> | SOURCE ONLY   | RN50         | 93.3 | 88.5 | 78.8   | 86.9   | 38M     |
> | DANN          | RN50         |  94.5 | 91.5 | 79.0   | 88.3   | 38M     |
> | CDAN          | RN50         |  94.7 | 92.0 | 79.0   | 88.6   | 38M     |
> | SOURCE ONLY   | CLIP-RN50    | 93.0 | 90.7 | 78.7   | 87.4   | 102M    |
> | DANN          | CLIP-RN50    |  95.0 | 91.7 | 79.2   | 88.6   | 102M    |
> | CDAN          | CLIP-RN50    |  93.7 | 93.0 | 80.0   | 88.9   | 102M    |
> | DAN | ResNet50     | 93.3 |  92.2 |  77.6 |  87.7 | 48.9M   |
> | D-CORAL  |  ResNet50     | 93.6  | 91.7  | 77.1  | 87.5 | 47.5M |
> | DANN          | ResNet50     | 95.7 | 91.8 | 77.9   | 87.8   | 48.9M   |
> | PGA           | Prompt-tuning|  96.8 | 95.7 | 84.6   | 92.4   | 114k    |
> | MPGA          | Prompt-tuning|  97.4 | 96.5 | 84.7   | 92.9   | 131k    |
>
> Specifically, we applied DANN and CDAN on CLIP’s ResNet50 configured in three different ways: with a randomly initialized fully connected classifier (RN50-FC), with a frozen text encoder (RN50), and using the entire CLIP backbone (CLIP-RN50). The results, presented in the table below, demonstrate that appropriately adapting prior UDA methods to different parts of the CLIP model can yield better results compared to traditional methods on a ResNet backbone (e.g., DAN, CORAL, DANN). Eventhough, PGA and MPGA still exhibit superior performance, even with significantly fewer parameters finetuned. We hypothesize that relying solely on source classification loss and another objective for invariant feature learning can degrade CLIP’s rich semantic representation [R8, R9, R10, R11], which is crucial for predicting target domain data. To counteract this, utilizing target pseudo data (similar to self-training baseline in Table 1) or adopting a more carefully-designed optimization procedure that better leverages information from both source and target data—similar to our proposed method—could enhance performance.
>
> We hope this response provides further insights into why traditional UDA methods may not perform as well as other prompt-based baselines. If the reviewer found any further unaddressed concerns, we are always happy to provide further clarifications and improve our work based on the constructive feedback from the reviewers.
>
> [R8] Kumar, Ananya, et al. "Fine-Tuning can Distort Pretrained Features and Underperform Out-of-Distribution." ICLR 22.
>
> [R9] Zheng, Zangwei, et al. "Preventing zero-shot transfer degradation in continual learning of vision-language models." ICCV 23.
>
> [R10] Lai, Zhengfeng, et al. "Padclip: Pseudo-labeling with adaptive debiasing in clip for unsupervised domain adaptation." ICCV 23.
>
> [R11] Ding, Yuxuan, et al. "Don't stop learning: Towards continual learning for the clip model." arXiv 22.

---

> > ### Comment · Reviewer_dhjM · 2024-08-11
> >
> > Thank you for the response about the additional experiments with traditional UDA. These results fully addressed my last concern. I also appreciate the fact that the authors have done these experiments and reported the results that will be very meaningful to subsequent reseaches. Since all my concerns have now been addressed, I have raised my score to 7.

---

> ### Author Response · Authors · 2024-08-11
> **Thank you**
>
> We sincerely thank the reviewer for the engaging discussion and greatly appreciate the valuable suggestions and feedback provided. We are honored by your appreciation of our paper and grateful for your support toward its acceptance. Thank you for your positive evaluation!

---

### Official Review · Reviewer_JYym · 2024-07-11

**Soundness:** 3
**Presentation:** 2
**Contribution:** 2
**Rating:** 4
**Confidence:** 3

**Summary:**

This paper proposes a novel approach called Prompt Gradient Alignment (PGA) for unsupervised domain adaptation (UDA). The key contributions are: (1) Formulating UDA as a multi-objective optimization problem with objectives for source and target domains; (2) Aligning gradients between objectives to encourage consensus; (3) Penalizing gradient norms to improve generalization; (4) Leveraging pre-trained vision-language models through prompt learning. The method is evaluated on standard UDA benchmarks and demonstrates consistent improvements over other prompt-based baselines. A theoretical generalization bound is also provided to justify the approach.

**Strengths:**

The strong empirical results across multiple UDA benchmarks are particularly impressive. The consistent outperformance of other prompt-based methods on datasets like ImageCLEF, Office-Home, and DomainNet (as shown in Tables 3-5) provides robust evidence for the method's effectiveness. The performance gains are substantial in many cases, with improvements of up to 4% on average accuracy compared to state-of-the-art methods.
The theoretical analysis offering a generalization bound (Section 4.5) adds credibility to the approach, providing insights into why the method works and under what conditions it can be expected to perform well. This combination of empirical success and theoretical grounding is a significant strength of the paper.
The ablation studies (Section 5.4, Table 6) effectively demonstrate the contribution of each component of the proposed method, providing a clear understanding of how different elements (like gradient alignment and norm penalization) contribute to the overall performance. The visualization of gradient similarity evolution (Figure 7) offers additional insights into the method's behavior during training.

**Weaknesses:**

The computational complexity and training time comparisons to baseline methods are notably absent. Without this information, it's difficult to assess the practical trade-offs of implementing PGA compared to existing methods. This is particularly important given the method's use of gradient manipulation, which could potentially increase computational requirements.
The paper's heavy reliance on pre-trained CLIP models raises questions about the method's applicability in scenarios where such pre-training is not available or suitable. While the use of pre-trained models is a strength in many cases, it could also be a limitation in domains significantly different from CLIP's training data.
The discussion of hyperparameter sensitivity is limited. Given the importance of hyperparameters like ρga and ρgn in the gradient alignment and norm penalization processes, a more thorough exploration of their impact on performance would be valuable.
While the method shows improvements over existing approaches, the degree of novelty is somewhat incremental. The core ideas build heavily on existing prompt-based and gradient manipulation techniques, which may limit the paper's impact.

**Questions:**

1. How does PGA's computational complexity compare to baseline methods? Are there significant differences in training time?
2. Have you explored performance on data types beyond image classification? How generalizable do you expect the method to be?
3. How sensitive is the method to hyperparameter choices, particularly for gradient alignment and norm penalization?
4. Given the reliance on CLIP pre-training, how well do you expect the method to perform in domains very different from CLIP's training data?
5. How does the method's performance change as the degree of domain shift varies? Is there a point where simpler approaches become competitive?

**Limitations:**

The reliance on CLIP pre-training potentially limits applicability to domains very different from CLIP's training data. It's unclear how well the method would perform on specialized domains (e.g., medical imaging, satellite imagery) that are not well-represented in CLIP's training set.
The paper does not thoroughly explore potential failure cases or scenarios where the method might struggle. For instance, how does the method perform when there are extreme label distribution shifts between source and target domains? Or when the domain shift is particularly severe?
The scalability of the method to very large datasets or high-dimensional data is not addressed. It's unclear how the computational requirements scale with dataset size or feature dimensionality.

---

> ### Author Rebuttal · Authors · 2024-08-06
>
> 1. In Figure 1 of the attached PDF we provide the complexity for some comparative baselines. Accuracy curve (left): While DANN and CDAN obtain their best performance at approximately 77% after more than 1000s, PGA and MPGA  achieve 84% within 100s. Besides, the first stage of pairwise source-target training of MPA takes 159s, followed by 35s for the second stage to actually train the final model. Number of Trainable Parameters (middle): PGA and MPGA, with fewer than 140k parameters, require significantly fewer parameters than MPA, DANN and CDAN, which have around 1M, 48.9M and 51.7M parameters, respectively. GPU Memory Usage (right)} PGA, MPGA, and MPA exhibit substantially lower memory footprints, around 1300MB compared to 7000MB of DANN and CDAN throughout training.
>
> 2. Thank you for the suggestion. Given the task-agnostic nature of our method, where PGA/MPGA utilizes only gradient information, and the successful application of CLIP-based methods in diverse tasks like image segmentation [R2] and object detection [R3], we believe our approach could be similarly effective in other tasks. For instance, applying our method to image segmentation, which requires predictions for each pixel, would necessitate training an additional decoder. Due to constraints in the rebuttal period, this extension is not straightforward, and is considered beyond the scope of our current experimental setup. We leave this for future work. Nonetheless, we believe that our method's superior results in image classification are sufficient to verify its benefits, as we adhere closely to the standard protocols of related UDA methods.
>
> 3. Figure 2 in the attached PDF shows that PGA is generally not sensitive to $\rho_{ga}$ and $\rho_{gn}$ within their acceptable range, i.e. 1e-2 to 10 for $\rho_{ga}$ and 1e-5 to 0.1 for $\rho_{gn}$. Specifically, (i) a too large value of $\rho_{gn}$ is less effective than smaller ones; (ii) ImageCLEF prefers larger values of $\rho_{ga}$ while OfficeHome prefers smaller ones, suggesting that source and target domains in the former dataset may be more similar than those in the latter, hence over-matching gradients in the latter dataset may be adverse.
> 4. It is true that when source and target domains are very different  from CLIP's training data, prompt learning alone may not be sufficient to adapt well to the target domain. Yet, our method is still expected to perform better than prompt-based baselines. A failure case is discussed in our adaptation to the QuickDraw domain of DomainNet, which contains black and white sketches. Since this domain differs significantly from CLIP's training data, all prompt-based methods fail to achieve good accuracy and perform worse than some full-finetuning methods. This observation suggests integrating our methods with other invariant feature learning methods (lines 32-33 in our manuscript) or finetuning more parameters can better adapt to the target domain. Nonetheless, our PGA and MPGA still yield the highest results among prompt-based ones, indicating that our gradient alignment and norm penalization procedure is more helpful in closing domain gap than previous DAPL and MPA. Another example where prompt-based cannot surpass traditional UDA baselines is Rw→Cl from OfficeHome in Table 5.
>
>     To conclude, for specialized domains not well-represented by the pretrained CLIP model, a more effective strategy might involve updating additional parameters, such as the layers close to the output in both vision and text encoders, beyond just the prompts. This could lead to better domain adaptation at the expense of computational complexity increase. We acknowledge this limitation in Appendix D and leave it for future work.
>
> 5. We test PGA on the setting of label shift following [R1], where the source or target domains are down-sampled with only 30% of data from the first-half of the classes are taken (indicated by **s-** prefix). Results presented in Table 4 in the attached PDF, table below and Table 5 in main text show the effectiveness of PGA on different levels of label shift.
> | Method      | Ar→sCl | Ar→sPr | Ar→sRw | Cl→sAr | Cl→sPr | Cl→sRw | Pr→sAr | Pr→sCl | Pr→sRw | Rw→sAr | Rw→sCl | Rw→sPr | Avg  |
> |-|-|-|-|-|-|-|-|-|-|-|-|-|-|
> | ResNet-50   | 41.5  | 65.8  | 73.6  | 52.2  | 59.5  | 63.6  | 51.5  | 36.4  | 71.3  | 65.2  | 42.8  | 75.4  | 58.2 |
> | DANN        | 47.8  | 55.9  | 66.0  | 45.3  | 54.8  | 56.8  | 49.4  | 48.0  | 70.2  | 65.4  | 55.5  | 72.7  | 58.3 |
> | JAN         | 45.8  | 69.7  | 74.9  | 53.9  | 63.2  | 65.0  | 56    | 42.5  | 74    | 65.9  | 47.4  | 78.8  | 61.4 |
> | CDAN        | 51.1  | 69.7  | 74.6  | 56.9  | 60.4  | 64.6  | 57.2  | 45.5  | 75.6  | 68.5  | 52.7  | 79.8  | 63.0 |
> | IWDANN      | 48.7  | 62.0  | 71.6  | 50.4  | 57.0  | 60.3  | 51.4  | 41.1  | 69.9  | 62.6  | 51.0  | 77.2  | 58.6 |
> | IWJAN       | 44.0  | 71.9  | 75.1  | 55.2  | 65.0  | 67.7  | 57.1  | 42.4  | 74.9  | 66.1  | 46.1  | 78.5  | 62.0 |
> | IWCDAN      | 52.3  | 72.2  | 76.3  | 56.9  | 67.3  | 67.7  | 57.2  | 44.8  | 77.8  | 67.3  | 53.3  | 80.6  | 64.6 |
> | PCT| 57.5 | 78.2 | 80.5  | 66.7  | 74.3  | 75.4  | 64.6  | 50.7  | 81.3  | 72.9  | 57.3  | 83.5  | 70.2 |
> | PGA (Ours)  | 57.4  | 84.8  | 86.4  | 76.0  | 84.6  | 85.6  | 74.5  | 57.1  | 86.1  | 75.9  | 57.4  | 85.3  | 75.9 |
> 6. Scalability w.r.t dataset-size: Our experiments already included small (ImageCLEF), medium (OfficeHome), and large-scale (DomainNet, S2RDA) datasets. Furthermore, our proposed optimization procedure is not data-dependent, hence the increase in data size or dimension would not cause computational overhead for PGA and MPGA compared to other baselines.
>
> [R1] Tanwisuth, Korawat, et al. "A prototype-oriented framework for unsupervised domain adaptation." NeurIPS 21
>
> [R2] Wang, Zhaoqing, et al. "Cris: Clip-driven referring image segmentation." CVPR 22.
>
> [R3] Wu, Xiaoshi, et al. "Cora: Adapting clip for open-vocabulary detection with region prompting and anchor pre-matching." ICCV 23.

---

> ### Comment · Area_Chair_27fr · 2024-08-13
>
> Hi,
>
> Could you take a look at the authors rebuttal and finalize your rating?
>
> Thanks, AC

---

### Official Review · Reviewer_YFTB · 2024-07-12

**Soundness:** 3
**Presentation:** 3
**Contribution:** 3
**Rating:** 8
**Confidence:** 4

**Summary:**

To enhance both transferability and discriminability for prompt learning based domain adaptation, this paper proposes a Prompt Gradient Alignment (PGA) method. PGA encompasses multiple domain-wise classification objectives, cosine similarity maximization regularizers between prompt gradients of different domains, and prompt gradient norm penalty for each classification objective. It can thus achieve both inter-domain gradient alignment and flat minima enforcement. To efficiently and effectively update the designed prompts, a practical gradient update procedure is devised and works under both single-source and multi-source UDA. Empirical results shown the superiority of the proposed method.

**Strengths:**

1. A novel method with an information-theoretic explanation. Aligning prompt gradients across domains moves the shared prompt towards the low-loss region of both domains, such that domain-invariant class-discriminative features can be captured, thus benifiting both domains.
2. A practical, flexible, and efficient gradient update procedure. It has a weighting term on the source signal to control how much emphasis should be put on the target domain and avoids the costly Hessian computation.
3. A well-written article. The paper has a smooth logic and is easy to understand.

**Weaknesses:**

1. The abstract should reflect the technical highlights. For example, UDA is essentially a multiple-objective optimization problem, which is not the technical novelty of this paper.

2. As indicated by Theorem 4.1, the proposed method minimizing source empirical loss, gradient norms, and gradient misalignment, model and reduce the upper bound of the established generalization error. Nevertheless, it is not intuitive why minimizing gradient norms and gradient misalignment are beneficial for the performance on the target domain.

3. In Theorem 4.1, g_t^src and g_t^T are the gradients w.r.t. P_t of source loss and target loss respectively. The two gradients should be derived from the same prompt or different prompts? Can the gradient alignment theory be extended to other spaces like image, feature, and output spaces?

4. Due to use of pseudo labels for the target domain, though they are improved by confidence filtering, the accuracy of pseudo labels may greatly affect the success of inter-domain gradient alignment. It is encouraged to investigate the relationship between them.

5. Some important related works are missing in the discussion of related works, e.g., [a].

[a] Tang et al. Unsupervised domain adaptation via distilled discriminative clustering. PR, 2022.

6. Validation on larger datasets like S2RDA [b] is important for three reasons. First, it is a more large-scale, realistic, challenging multi-domain benchmark dataset. Second, it is tailored for synthetic-to-real transfer, more in line with industrial needs. Third, this paper leverages the powerful generalization capability of large-scale pre-trained vision-language models and fine-tuning with a large amount of training data can avoid the potential risk of overfitting.

[b] Tang and Jia. A New Benchmark: On the Utility of Synthetic Data with Blender for Bare Supervised Learning and Downstream Domain Adaptation. CVPR, 2023.

7. What circumstances does the overfitting issue appear? Why does penalizing gradient norms alleviate the overfitting issue? Are there any theoretical reasons?

8. In Eq. (6), L_tgt should be L_T? In Eq. (20), L_s should be L_s^PGA? The authors may re-write the final overall objective for clarity.

**Questions:**

See Weaknesses.

---

> ### Author Rebuttal · Authors · 2024-08-06
>
> 1. Thanks for the suggestion, we will revise the abstract to reflect our technical contributions more clearly. Regarding the multiple objective optimization viewpoint, our MOO problem consists of per-domain objectives, motivated by the strong performance of the self-training baseline in Section 1. This empirical results show that optimizing target loss using pseudo data alone is already a strong baseline, which is not typically found in traditional UDA methods. They instead often combine source loss with auxiliary domain alignment objectives.
> 2.
> - As can be seen from the first term on the R.H.S of Theorem 4.1, minimizing these two terms will reduce the generalization error, thereby closing the gap between target population risk and source empirical risk. Furthermore, as we also minimize the latter through the cross entropy loss on labeled source data, the former will be reduced. This is the ultimate goal an UDA method aims to achieve, as low population target risk indicates a model having a good generalizability on target data.
> - Intuitively, maximizing gradient alignment encourages the optimization trajectories to be the same for all domains, and minimizing gradient norm helps find flat regions for both shared and domain-specific prompts, which alleviates overfitting and generally leads to better performance on source/target data [4, 18, 60]. Altogether, performance on target domain can be improved.
> 3.
> - ${g_t^{src};g_t^{tgt}}$  should be derived from the same prompt, and in the theorem, to ease the proof, they refer to the whole prompt set $[P_{sh}, P_S, P_T]$ at time step $t$. Note that this does not contradict our gradient alignment strategy which is applied on the shared prompt only. Indeed, as we showed in Remark A.7, since $P_T$ is not involved when computing the source loss, we can write $g^{src}_t = [g^{sh,src}_t, g^S_t, 0]$. Similarly, $g^{tgt}_t = [g^{sh, tgt}_t, 0, g^T_t]$. Therefore, the inner product between $g^{src}_t$ and $g^{tgt}_t$ is equal to the inner product between $g^{sh,src}_t$ and $g^{sh,tgt}_T$.
>  - Extension of the theory to other spaces like image, feature, and output spaces: Please note that the second term in Theorem 4.1, which contains the KL-divergence between the two underlying source and target distributions $\mu, \mu'$, is the motivation for aligning these two distributions in image, feature or output space. Normally, this is done in feature space due to the rich representation captured by the feature extractor. For example, marginal feature alignment is adopted in [R1] and conditional feature alignment in [R2]. In the case of conditional alignment, one has to assign pseudo labels to target data using the learned source model [R3], which may lead to accumulation error. In our case, as we use zero-shot CLIP to predict pseudo labels, the error could be alleviated. Last but not least,  as mentioned in the paper, incorporating our gradient matching and norm penalization into those methods fully reflects the two terms from the R.H.S of the bound, hence could further boost performance.
> 4. We provide an ablation varying the value of threshold $\tau$ in Table 2 of the attached PDF. Our methods are not sensitive to $\tau$, and the best result is obtained at a reasonable trade-off between the quantity and quality of pseudo data.
> 5. Thank you for your suggestion, we will add this paper to the related works and its results to the Office-Home experiment, where they obtain an average accuracy of 71.4 compared to 75.8 (ours).
> 6. We include the performance of CLIP-based models on two Synthetic-to-Real datasets S2RDA-49 and S2RDA-MS-39 in Table 1 of the attached PDF. PGA achieves the best performance among the baselines.
> 7.
> - When does the overfitting issue appear: Since we are optimizing multiple objectives across domains, naively minimizing them might lead to overfitting on some particularly easy-to-optimize or limited-data objectives, while forcibly minimizing them could harm other objectives.
> - Why does penalizing gradient norms help: Briefly, in standard supervised training, minimizing gradnorm will lower the population risk, hence reduce overfitting. Similarly, in UDA, applying this on source and target data will lead to lower population risks on respective domains. Specifically, followed Theorem 1 in [55], we can upper bound these risks as:
> $[L_{\mu}(P), L_{\mu'}(P)] \leq \max_{||\epsilon_{sh}||\leq\rho_{gn}}[\max_{||\epsilon_S||\leq\rho_{gn}}L_S(P_{sh}+\epsilon_{sh},P_{S}+\epsilon_S)+f_S(||P||), \max_{||\epsilon_T||\leq\rho_{gn}}L_T(P_{sh}+\epsilon_{sh},P_{T}+\epsilon_T)+f_T(||P||)],$
>     where $f_S,f_T$ are strictly increasing functions.
> Furthermore, the worst-case source loss can be approximated as
> $\max_{||\epsilon_{sh}||\leq\rho_{gn}}\max_{||\epsilon_S||\leq\rho_{gn}}L_S(P_{sh}+\epsilon_{sh},P_{S}+\epsilon_S) \approx L_S(P)+\rho_{gn}(||g_{sh,S}||+||g_S||),$ and similarly for the worst-case target loss.
>
> 8.
> - "In Eq. (6)". Thanks for pointing this out. Yes, $L_T$ is the correct notation. Also in the definition of the gradient w.r.t target loss (in line 224), the correct notation should be $g_t^{tgt}$. We will fix these in the revision.
> - In Eq. (20): In fact, it should be $L_S$, which is similar to $L_T$ in the approximation of Eq.(19).
> Informally, the PGA gradients corresponding to the source objective are the derivative of $L_S^{\text{PGA}}$ w.r.t $P$, which are then approximated by the derivative of the source loss at  $L_S$ the perturbed prompts: $P_{sh}=P_{sh}-\rho_{ga}{b}+\rho_{gn}\frac{g_{sh,S}}{||g_{sh,S}||}, P_S =P_S+\rho_{gn}\frac{g_S}{||g_S||}$. (This approximation follows the derivation of Eq.(13)).
> - Overall objective: please refer to the general response.
>
> [R1] Nguyen, A. Tuan, et al. "KL guided domain adaptation.", ICLR 22
>
> [R2] Nguyen, A. Tuan, et al. "Domain invariant representation learning with domain density transformations.", NeurIPS 21
>
> [R3] Long, M., et al. "Conditional adversarial domain adaptation", NeurIPS 18

---

> > ### Comment · Reviewer_YFTB · 2024-08-12
> >
> > I'd like to acknowledge receipt of your response, e.g., the supplementary experiments involving traditional UDA and larger datasets. The results effectively alleviate my previous apprehensions. I commend the authors for conducting these experiments and sharing results that will significantly impact future research. Given that all my concerns have been satisfactorily addressed, I have revised my rating to 8.

---

> > > ### Author Response · Authors · 2024-08-12
> > > **Thank you**
> > >
> > > Thank you for raising your concerns, and for your detailed review and insightful suggestions. Addressing these points will undoubtedly enhance the paper's presentation and more thoroughly examine our proposed method. We will further follow your valuable recommendation.

---

### Author Rebuttal · Authors · 2024-08-06

We thank all reviewers for the valuable and supportive feedback. We appreciate that our paper is recognized for having strong **empirical results** (reviewers JYym, dhjM), a **generalization bound** which adds credibility to our approach (all reviewers), a **novel** method with **clear intuitions** (reviewer aPJr, dhjM, YFTB) and efficiency (reviewer YFTB, dhjM), and a **clear and coherent structure** (reviewer aPJr, YFTB).

In this paper, we tackle the unsupervised domain adaptation (UDA) problem by framing it as a multi-objective optimization task, which is motivated by the strong performance of self-training on the target domain, i.e. minimizing target domain loss on pseudo-labeled data. Furthermore, to promote inherent consensus between domains and encourage model flatness, we propose to align their gradients and penalize their gradnorms with an efficient gradient update procedure, supported by a backup generalization error bound.

Inspired by the reviewers' comments, we have enhanced our manuscript by including additional experimental results, responses, and clarifications, which are briefly summarized as follows:
- Results on S2RDA datasets (reviewer YFTB), ViT backbones (reviewer aPJr), and label-shift setting (reviewer JYym)
- Analysis on hyper-parameter sensitivity (reviewers JYym, YFTB) & computational complexity (reviewer JYym)
- Clarification on the final loss function (reviewers YFTB, dhjM)
- Discussion on failure cases (reviewer JYym) and on the difference between our method and gradient-surgery ones (reviewer aPJr).

We also attach a **PDF** file that contains our main additional results, which we will incorporate in the revision. Except for the complexity benchmark, results for all baselines are directly reported from prior work for a fair comparison. We hope that the new results will help address reviewers' concerns.

Here we would like to clarify the formula for final loss functions, and then present detailed responses separately for each reviewer.

**Final loss function**
As we cast UDA as a MOO problem, the ideal final objectives, in the case of single UDA, would be  $[L_S^{PGA}(P), L_T^{PGA}(P)]$, where:

$L_T^{PGA}(P) := L_T(P_{sh} - \rho_{ga}\frac{g_{sh,S}}{||g_{sh,S}||.||g_{sh,T}||} +  \rho_{gn}\frac{g_{sh,T}}{||g_{sh,T}||}, P_T + \rho_{gn}\frac{g_T}{||g_T||}),$

$L_S^{PGA}(P) := L_{S}(P_{sh} - \rho_{ga}\frac{g_{sh,T}}{||g_{sh,S}||.||g_{sh,T}||} +  \rho_{gn}\frac{g_{sh,S}}{||g_{sh,S}||}, P_S + \rho_{gn}\frac{g_S}{||g_S||}).$

As mentioned in the paper (e.g. line 58-59, 172-174), we use scalarization method [30], i.e. simply reweighting loss functions with $\lambda$ put on the PGA source objective. As a result, the gradient updates for prompts are

$g_{sh,T}^{PGA},g_T^{PGA} = \nabla_P L_T^{PGA}(P), \quad g_{sh,S}^{PGA},  g_S^{PGA} = \nabla_P L_S^{PGA}(P),$

$P_S = P_S-\eta g ^{PGA}_S, P_T = P_T-\eta g^{PGA}_T,$

 $P_{sh} = P_{sh} - \eta (g_{sh,T}^{PGA}+\lambda g_{sh,S}^{PGA}).$

However, computing these PGA gradients will trigger the computation of Hessian matrix. Hence, we implicitly calculate them via a practical algorithm that is shown in Eq.(19, 20) in the main text.

---

### Decision · Program_Chairs · 2024-09-25

**Decision:**

Accept (poster)

**Comment:**

This paper proposes an approach for enhancing domain generalization by aligning prompt gradient. While reviewers believe that the paper is well-written and the approach is novely, concerns are raised in terms of the experiments and comparisons. The rebuttal successfully addressed most of the concerns, while there are still doubts in terms of whether the approach can be used in traditional DA approaches. Overall, the AC believes the paper provides a new pespective for DA research and thus recommends acceptance.